# Geometric Origin of the Galaxies' Dark Side

**Leonardo Modesto \*, Tian Zhou**  **and Qiang Li**

Department of Physics, Southern University of Science and Technology, Shenzhen 518055, China; zhout6@mail.sustech.edu.cn (T.Z.)
\* Correspondence: lmodesto@sustech.edu.cn

**Abstract:** We show that Einstein's conformal gravity can explain, simply, and on the geometric ground, galactic rotation curves, without the need to introduce any modification in both the gravitational as well as in the matter sector of the theory. The geometry of each galaxy is described by a metric obtained, making a singular rescaling of Schwarzschild's spacetime. The new exact solution, asymptotically anti-de Sitter, manifests an unattainable singularity at infinity that cannot be reached in finite proper time; namely, the spacetime is geodetically complete. It deserves to be noticed that, in this paper, we have a different opinion from the usual one. Indeed, instead of making the metric singularity-free, we make it apparently but harmlessly even more singular than Schwarzschild's. Finally, it is crucial to point out that Weyl's conformal symmetry is spontaneously broken into the new singular vacuum rather than the asymptotically flat Schwarzschild's one. The metric is unique according to the null energy condition, the zero acceleration for photons in the Newtonian regime, and the homogeneity of the Universe at large scales. Once the matter is conformally coupled to gravity, the orbital velocity for a probe star in the galaxy turns out to be asymptotically constant consistent with the observations and the Tully–Fisher relation. Therefore, we compare our model with a sample of 175 galaxies and show that our velocity profile very well interpolates the galactic rotation curves after a proper choice of the only free parameter in the metric. The mass-to-luminosity ratios of galaxies turn out to be close to 1, consistent with the absence of dark matter.

**Keywords:** galactic rotation curves; dark matter; conformal gravity

## 1. Introduction

Despite the enormous successes of Einstein's theory of gravity, it appears to be about "twenty five percent wrong". To date, the scientists proposed two possible solutions to the problems that are known under the name of "dark matter" or "dark gravity", and both are extensions of Einstein's field equations. The first proposal consists of modifying the right side of Einstein's equations, while according to the second proposal, are modified on the left-hand side. Indeed, to take into account all the observational evidence—galactic rotation curves, structure formation in the universe, CMB spectrum, bullet cluster, and gravitational lensing—it seems necessary to somehow modify Einstein's field equations. However, in this paper, we propose the following different approach, namely: "understand gravity instead of modifying it".

In this document, we do not pretend to provide a definitive answer to the "mystery of missing mass" or "missing gravity in the universe", but we only focus on the galactic rotation curves. Nevertheless, we believe our result to be quite astonishing on both the theoretical and observational sides.

The analysis here reported, which follows the previous paper [1][1], is universal and applies to any conformally invariant theory, nonlocal [2–5], or local [6], that has the Schwarzschild metric as an exact [7] and stable solution [8–12].

However, for the sake of simplicity, we will focus on Einstein's conformal gravity, whose general covariant action functional [13–21] reads

$$S = \int d^4x \sqrt{-\hat{g}} \left( \phi^2 \hat{R} + 6\hat{g}^{\mu\nu} \partial_\mu \phi \partial_\nu \phi - 2h\phi^4 \right), \tag{1}$$

which is defined on a pseudo-Riemannian spacetime Manifold $\mathcal{M}$ equipped with a metric tensor field $\hat{g}_{\mu\nu}$, a scalar field $\phi$ (the dilaton), and it is invariant under the following Weyl conformal transformation:

$$\hat{g}'_{\mu\nu} = \Omega^2 \hat{g}_{\mu\nu}, \quad \phi' = \Omega^{-1} \phi, \tag{2}$$

where $\Omega(x)$ is a general local function. In (1), $h$ is a dimensionless constant that has to be selected extremely small to have a cosmological constant compatible with the observed value. However, we here assume $h = 0$ because the presence of a tiny cosmological constant will not affect our result (see Appendix B for more details). For completeness and to show the exactness of the solutions that we will expand later on, we here remind the reader of the equations of motion for the theory (1) for $h = 0$,

$$\phi^2 \hat{G}_{\mu\nu} = \hat{\nabla}_\nu \partial_\mu \phi^2 - \hat{g}_{\mu\nu} \hat{\Box} \phi^2 - 6 \left( \partial_\mu \phi \partial_\nu \phi - \frac{1}{2} \hat{g}_{\mu\nu} g^{\alpha\beta} \partial_\alpha \phi \partial_\beta \phi \right),$$
$$\hat{\Box} \phi = \frac{1}{6} \hat{R} \phi. \tag{3}$$

The Einstein–Hilbert action for gravity is recovered if the Weyl conformal invariance is broken spontaneously in exact analogy with the Higgs mechanism in the standard model of particle physics (for more details we refer the reader to [22]).

If the dimensionless parameter $h$ vanishes, the potential is identically zero everywhere and the vacuum solution is $\phi = $ const., with the latter constant proportional to Newton's constant. Therefore, the full vacuum will be consistent with: $(\phi = $ const., $\hat{R}_{\mu\nu} = 0)$, as required by General Relativity in the absence of the cosmological constant. On the other hand for $h \neq 0$, the vacuum solution is again $\phi = $ const., but now $\hat{R}_{\mu\nu} = \Lambda \hat{g}_{\mu\nu}$, namely the vacuum is: $(\phi = $ const., $\hat{R}_{\mu\nu} = \Lambda \hat{g}_{\mu\nu})$. Therefore, for $h \neq 0$ the vacuum is not Minkowski's spacetime, but the de Sitter one. Finally, according to (3) the vacuum $\phi = 0$ is degenerate because the two EoMs in (3) are identically satisfied for any metric $\hat{g}_{\mu\nu}$.

To end up with General Relativity after the conformal symmetry is spontaneously broken, the vacuum for the scalar field in the theory (1) (exact solution of the equations of motion (3)) must be $\phi = $ const. $= \kappa_4^{-1} = 1/\sqrt{16\pi G}$, together with the metric satisfying $R_{\mu\nu} \propto \hat{g}_{\mu\nu}$. Therefore, replacing $\phi = 1/\sqrt{16\pi G} + \varphi$ in the action (1) and using the conformal invariance to eliminate the gauge-dependent Goldstone's degree of freedom $\varphi$, we end up with Einstein–Hilbert's action in the presence of the cosmological constant,

$$S_{\text{EH}} = \frac{1}{16\pi G} \int d^4x \sqrt{-\hat{g}} \left( \hat{R} - 2\Lambda \right), \tag{4}$$

where $\Lambda$ is consistent with the observed value for a proper choice of the dimensionless parameter $h = 16\pi G \Lambda$ in the action (1). Ergo, Einstein's gravity is simply the theory (1) in the spontaneously broken phase of Weyl conformal invariance [22,23].

Let us now expand on the exact solutions in conformal gravity. Given the conformal invariance (2), any rescaling of the metric $\hat{g}_{\mu\nu}$ accompanied by a non-trivial profile for the dilaton field $\phi$ is also an exact solution, namely

$$\hat{g}^*_{\mu\nu} = Q^2(x) \hat{g}_{\mu\nu} \quad \phi^* = Q(x)^{-1} \phi, \tag{5}$$

solve the EoM obtained by varying the action (1) with respect to $\hat{g}_{\mu\nu}$ and $\phi$.

To date, the rescaling (5) has been used to show how the singularity issue disappearance in conformal gravity [22–25]. However, and contrary to the previous papers, we here

focus on a not asymptotically flat rescaling of the Schwarzschild metric as a workaround to the non-Newtonian galactic rotation curves. Moreover, the logic in this project is opposite to the one implemented in the past works and it is somehow anti-intuitive. In fact, in Section 2, instead of removing the spacetime's singularities, we deliberately introduce a singular function $Q(x)$ which leads to an unreachable asymptotic spacetime singularity. However, as it will be proved in Section 2, the spacetime stays geodetically complete. Indeed, the proper time to reach the singularity at the edge of the universe will turn out to be infinite.

Notice that to give a physical meaning to the metric (5), conformal symmetry has to be broken spontaneously to a particular vacuum specified by the function $Q(x)$. The uniqueness of such rescaling will be discussed in Section 3. In the spontaneously broken phase of conformal symmetry, observables are still invariant under diffeomorphisms.

In Section 4, we apply the spherically symmetric metric constructed in Section 2 to drive the orbital rotation velocity of probe particles. Then, in Section 5, the effective gravitational potential of a single star is obtained and we subsequently obtain the galactic rotation curves by summing up effective potential contributions from all stars in a galaxy. Afterward, in Section 7, we compare our galactic rotation velocity profile to the observation data of 175 galaxies, and meanwhile, determine the free parameters in our model by data fitting. The fitting results are shown in the Appendixes C and D. Finally, we conclude the above discussion and summarize the advantages of our models.

## 2. The Spherically Symmetric Solution in Conformal Gravity

As explained in the introduction, given an exact solution of Einstein's conformal gravity, any rescaled metric is an exact solution too, if the metric is accompanied by a non-trivial profile for the dilaton. Therefore, we here consider the following conformal rescaling of the Schwarzschild spacetime[2],

$$d\hat{s}^{*2} = Q^2(x)\left[-\left(1 - \frac{2GM}{x}\right)dt^2 + \frac{dx^2}{1 - \frac{2GM}{x}} + x^2(d\theta^2 + \sin^2\theta d\varphi^2)\right], \quad (6)$$

$$Q(x) = \frac{1}{1 - \frac{\gamma^*}{2}x}, \quad (7)$$

$$\phi^* = Q(x)^{-1}\kappa_4^{-1},$$

where we identified $x$ with the radial coordinate. The reason for the particular rescaling $Q(x)$ will be clarified shortly, making use of a more suitable radial coordinate. Notice that $Q(x)$ is singular for $x = 2/\gamma^*$, and, therefore, the metric is defined for $x < 2/\gamma^*$. However, we will prove in the next section that the asymptotic singularity is unattainable; namely, it requires an infinite amount of proper time to be reached. As a remnant of the previous work [1], we named $\gamma^*/2$ the free inverse length scale present in the solution. However, to manifestly identify the effect of the conformal symmetry it would be useful and more suitable to define: $\gamma^*/2 \equiv \ell_c$, which we will refer to as the characteristic scale of the system.

To show that the scaling factor $Q(x)$ in (7) is the only one compatible with (i) $g_{00} = -1/g_{11}$ (we will expand on the uniqueness of the metric in Section 3), we make a coordinate transformation to the usual radial Schwarzschild coordinate "$r$", which identifies the physical radius of the two-sphere (ii). The new radial coordinate $r$ is related to $x$ as follows,

$$x = \frac{r}{1 + \frac{\gamma^*}{2}r}, \quad (8)$$

$$\frac{\partial x(r)}{\partial r} = \frac{1}{Q(r)^2}, \quad (9)$$

and the metric turns into:

$$d\hat{s}^{*2} = -Q^2(r)\left(1 - \frac{2GMQ(r)}{r}\right)dt^2 + \frac{dr^2}{Q^2(r)\left(1 - \frac{2GMQ(r)}{r}\right)} + r^2(d\theta^2 + \sin^2\theta d\varphi^2),$$

$$Q(r) = 1 + \frac{\gamma^*}{2}r \quad \left(\text{notice that } x = \frac{r}{Q(r)}\right),$$
$$\phi^*(r) = Q(r)^{-1}\kappa_4^{-1}. \tag{10}$$

It deserves to be noted that any rescaling that differs from the one in (7) is not consistent with the two requirements above, namely (i) and (ii). Therefore, in the infinite class of exact solutions conformally equivalent to the Schwarzschild metric, there is only one geometry non-asymptotically flat consistent with $g_{00} = -1/g_{11}$ and two-dimensional transverse area $4\pi r^2$. Notice that $Q(r)$ in (10) is only linear in $r$, which is the minimal modification of the metric compatible with analyticity. As mentioned above, we will expand further on the uniqueness of the metric in Section (3).

### 2.1. Regularity of the Kretschmann and Weyl Square Invariants

As a first check of the regularity, we look at the spacetime in $x = 2/\gamma^*$. Since the Schwarzschild spacetime is Ricci flat, before the rescaling the first non-trivial curvature invariant is the Kretschmann scalar, which reads:

$$\hat{K} := \hat{R}_{\alpha\beta\gamma\delta}\hat{R}^{\alpha\beta\gamma\delta} = \hat{C}_{\alpha\beta\gamma\delta}\hat{C}^{\alpha\beta\gamma\delta} = \frac{48G^2M^2}{x^6}, \tag{11}$$

where in the last equality we used that $\hat{R}_{\alpha\beta} = 0$ and introduced the Weyl tensor $\hat{C}_{\alpha\beta\gamma\delta}$. Under the Weyl rescaling (2) the Weyl tensor, for the following position of the indexes, is invariant, namely

$$\hat{C}^{*\alpha}{}_{\beta\gamma\delta} = \hat{C}^{\alpha}{}_{\beta\gamma\delta}. \tag{12}$$

Hence, the Kretschmann scalar (11) for the metric (7) turns into:

$$\begin{aligned}
\hat{C}^{*2} &= \hat{C}^{*\alpha}{}_{\beta\gamma\delta}\,\hat{C}^{*\mu}{}_{\nu\rho\sigma}\,\hat{g}^*_{\alpha\mu}\hat{g}^{*\beta\nu}\hat{g}^{*\gamma\rho}\hat{g}^{*\delta\sigma} \\
&= \hat{C}^{\alpha}{}_{\beta\gamma\delta}\,\hat{C}^{\mu}{}_{\nu\rho\sigma}\,\hat{g}_{\alpha\mu}\hat{g}^{\beta\nu}\hat{g}^{\gamma\rho}\hat{g}^{\delta\sigma}Q^2(x)\,Q^{-2}(x)\,Q^{-2}(x)\,Q^{-2}(x) \\
&= \frac{\hat{C}^2}{Q^4(x)}.
\end{aligned} \tag{13}$$

Finally, for the metric (7) we find:

$$\hat{C}^{*2} = \frac{\hat{K}}{Q^4(x)} = \frac{48G^2M^2}{x^6}\left(1 - \frac{\gamma^*}{2}x\right)^4, \tag{14}$$

which is zero in the limit $x \to \gamma^*/2$. The latter point, as we will show explicitly in the next subsection, represents the spatial infinity for the metric (7), because nothing can reach such a point in finite proper time. Therefore, the curvature invariant approaches asymptotically zero.

Using the radial coordinate $r$ the curvature invariant $\hat{C}^{*2}$ turns into ($x = r/Q(r)$):

$$\hat{C}^{*2}(r) := \hat{C}^{*2}(x(r)) = \frac{\hat{C}^2(x(r))}{Q^4(x(r))} = \frac{48G^2M^2}{r^6}Q^6(r)\,Q^{-4}(r) = \frac{48G^2M^2}{r^6}\left(1 + \frac{\gamma^*}{2}r\right)^2, \tag{15}$$

which is now zero for $r \to +\infty$ according to the inverse coordinate transformation from $x$ to $r$, namely

$$r = \frac{x}{1 - \frac{\gamma^*}{2}x}, \tag{16}$$

which diverges to infinity for $x \to \gamma^*/2$.

On the other hand, the Kretschmann scalar for the metric (10) is:

$$
\hat{K}^* = \frac{1}{2r^6}\Big[ -2\gamma^{*2}GMr^3(16 + 12\gamma^* r + 3\gamma^{*2}r^2) + \gamma^{*2}r^4(16 + 12\gamma^* r + 3\gamma^{*2}r^2)
$$
$$
+ 4G^2M^2(24 + 16\gamma^* r + 8\gamma^{*2}r^2 + 4\gamma^{*3}r^3 + \gamma^{*4}r^4) \Big]. \tag{17}
$$

At large distance the Kretschmann invariant for the metric (10) tends to a constant $\hat{K}^* \to 3\gamma^{*4}/2$, which means that the metric (10) describes asymptotically a spacetime of constant curvature. Indeed, at large scales the metric (10) approaches the anti-de Sitter spacetime with scalar curvature $R \to -3\gamma^{*2}$ in the limit $r \to +\infty$.

Therefore, the two curvature invariants computed above, namely $\hat{C}^{*2}$ and $\hat{K}^*$, are asymptotically finite, and $x = 2/\gamma^*$ is not a curvature singularity. The latter point, as we will show explicitly in the next subsection, represents the spatial infinity for the metric (7) because nothing can reach such a point in finite proper time.

Although, in this paper, we are concerned with the spacetime far outside the event horizon (indeed all the probes in the galaxy are stars and not black holes), the reader may worry about the singularity at $x = 0$ or $r = 0$. However, the resolution of singularities has been rigorously dealt with in several previous articles [22,23] and the results found there can be exported directly to the metric (7). Indeed, it is sufficient to rescale the latter metric as explicitly performed in [22]. For completeness, starting from the metric (10), we here provide an explicit example of geodetically complete spacetime from short to large distances, namely

$$
d\hat{s}^{*2} = S(r)\left[ -Q^2(r)\left(1 - \frac{2GMQ(r)}{r}\right)dt^2 + \frac{dr^2}{Q^2(r)\left(1 - \frac{2GMQ(r)}{r}\right)} + r^2 d\Omega^{(2)} \right],
$$
$$
Q(r) = 1 + \frac{\gamma^*}{2}r \qquad \left(\text{notice that } x = \frac{r}{Q(r)}\right),
$$
$$
\phi^*(r) = S(r)^{-1/2}Q(r)^{-1}\kappa_4^{-1},
$$
$$
S(r) = 1 + \frac{L^4}{r^4}, \tag{18}
$$

where $L$ is a parameter with the dimension of length (for more details and observational constraints on $L$ see [22,24,27]).

Finally, we notice that the geometry (10) has the same Penrose diagram of the Schwarzschild black hole because a conformal rescaling cannot change the causality structure of the spacetime.

### 2.2. Geodetic Completion: Conformally Coupled Particles

For the sake of simplicity from now, in the paper, we will remove the label "*" from the metric and the dilaton field. Let us start with a conformally coupled particle whose action reads:

$$
S_{\text{cp}} = -\int \sqrt{-f^2\phi^2 \hat{g}_{\mu\nu}dx^\mu dx^\nu} = -\int \sqrt{-f^2\phi^2 \hat{g}_{\mu\nu}\frac{dx^\mu}{d\lambda}\frac{dx^\nu}{d\lambda}}\, d\lambda, \tag{19}
$$

where $f$ is a positive constant coupling strength, $\lambda$ is the world-line parameter, and $x^\mu(\lambda)$ is the trajectory of the particle[3]. From (19), the Lagrangian reads:

$$
L_{\text{cp}} = -\sqrt{-f^2\phi^2 \hat{g}_{\mu\nu}\dot{x}^\mu \dot{x}^\nu}, \tag{20}
$$

and the translation invariance in the time-like coordinate $t$ implies

$$\frac{\partial L_{cp}}{\partial \dot{t}} = -\frac{f^2 \phi^2 \hat{g}_{tt} \dot{t}}{L_{cp}} = \text{const.} \equiv -E, \tag{21}$$

therefore, the equation of motion $\dot{t}$ reads

$$\dot{t} = \frac{L_{cp} E}{f^2 \phi^2 \hat{g}_{tt}}. \tag{22}$$

Since we are interested in evaluating the proper time for the particle to reach the singularity of the universe located in $x = 2/\gamma^*$, we choose the proper time gauge, namely $\lambda = \tau$. Therefore, $E$ can be formally interpreted as the energy of the test particle. From the equation $\hat{g}_{\mu\nu} \dot{x}^\mu \dot{x}^\nu = -1$, we have $L_{cp} = -f\phi$ then the Equation (22) is given by

$$\dot{t} = -\frac{E}{f \phi \hat{g}_{tt}}. \tag{23}$$

Replacing $\dot{t}$ from (23) in the radial geodesic equation $g_{tt} \dot{t}^2 + g_{xx} \dot{x}^2 = -1$ and using the solution of the EOM for $\phi$, namely $\phi = Q^{-1} \kappa_4^{-1}$, we end up with the following first-order differential equation for $x(\tau)$, namely

$$Q(x)^4 \dot{x}^2 + Q(x)^2 \left(1 - \frac{2GM}{x}\right) - \frac{E^2 \kappa_4^2}{f^2} Q(x)^2 = 0, \tag{24}$$

or, introducing the dimensionless parameter $e^2 \equiv \frac{E^2 \kappa_4^2}{f^2}$,

$$Q(x)^2 \dot{x}^2 = \frac{2GM}{x} + e^2 - 1. \tag{25}$$

Since we are interested in investigating the asymptotic completeness of the spacetime for large $x$, we can assume $x \gg 2GM$ and (25) simplifies to

$$Q(x)^2 \dot{x}^2 \simeq e^2 - 1, \tag{26}$$

which must be positive because $Q(x)^2 \dot{x}^2$ is surely positive. Replacing $Q(x)$ from (7) into (26) we obtain:

$$\frac{|\dot{x}|}{|1 - \frac{\gamma^*}{2} x|} \simeq \sqrt{e^2 - 1} > 0. \tag{27}$$

We here would like to study a particle moving from smaller to larger values of $x$, then $\dot{x} > 0$, moreover, $x < \gamma^*/2$, therefore (27) simplifies to

$$\frac{\dot{x}}{1 - \frac{\gamma^*}{2} x} \simeq \sqrt{e^2 - 1} > 0, \tag{28}$$

and the solution is:

$$\tau = \frac{2}{\gamma^* \sqrt{e^2 - 1}} \log\left(\frac{1 - \frac{\gamma^*}{2} x_0}{1 - \frac{\gamma^*}{2} x}\right) \quad \text{or} \quad x(\tau) = \frac{2}{\gamma^*}\left[1 - e^{-\frac{\gamma \tau}{2}}\left(1 - \frac{\gamma^*}{2} x_0\right)\right]. \tag{29}$$

According to the solution (29), the proper time to reach the edge of the universe located in $x = 2/\gamma^* = \ell_c$ is infinity. Moving to the radial coordinate $r$ defined in (8),

$$\lim_{x \to \frac{\gamma^*}{2}} r = \lim_{x \to \frac{\gamma^*}{2}} \frac{x}{1 - \frac{\gamma^*}{2} x} = +\infty. \tag{30}$$

Therefore, a massive particle will reach $r = +\infty$ in an infinite amount of proper time. Indeed, in the coordinate $r$, the radial geodesic Equation (25) turns into:

$$
\begin{aligned}
Q(r)^2 \dot{r}^2 \left( \frac{\partial x[r(\tau)]}{\partial r} \right)^2 &= \frac{2GMQ(r)}{r} + e^2 - 1, \\
\frac{\dot{r}^2}{Q(r)^2} &= \frac{2GMQ(r)}{r} + e^2 - 1
\end{aligned}
\tag{31}
$$

where we used (8) and (9),

$$
\begin{aligned}
\frac{\dot{r}}{Q(r)} &\simeq \sqrt{GM\gamma^* + e^2 - 1} := \mathrm{c} \ \text{ for } \ r \gg 2GM, \ \dot{r} > 0, \ \text{ and } \ GM\gamma^* + e^2 - 1 > 0, \\
\tau &= \frac{2}{\mathrm{c}\,\gamma^*} \log \left( \frac{1 + \dfrac{\gamma^*}{2} r}{1 + \dfrac{\gamma^*}{2} r_0} \right).
\end{aligned}
\tag{32}
$$

So far, we found that the proper time for a particle (conformally coupled) to reach the edge of the Universe is infinite in both $x$ and $r$ coordinates, in the former case the boundary is located at the finite value $x = \gamma^*/2$, in the latter case it is located in $r = +\infty$. In the next section, we will study the geodesic motion of massless particles.

*2.3. Geodetic Completion: Massless Particles*

For massless particles, the correct action, which is invariant under reparametrizations of the world line, $p' = f(p)$, is

$$
S_\gamma = \int \mathcal{L}_\gamma d\lambda = \int e(p)^{-1} \phi^2 \hat{g}_{\mu\nu} \frac{dx^\mu}{dp} \frac{dx^\nu}{dp} dp,
\tag{33}
$$

where $e(p)$ is an auxiliary field that transforms as $e'(p')^{-1} = e(p)^{-1}(dp'/dp)$ in order to guarantee the invariance of the action. The action (33) is not only invariant under general coordinate transformations but also the Weyl conformal rescaling (2).

The variation $\delta S_\gamma / \delta e$ gives

$$
d\hat{s}^2 = \hat{g}_{\mu\nu} dx^\mu dx^\nu = 0,
\tag{34}
$$

which is equivalent to saying that massless particles travel along the light cone.

The variation with respect to $x^\mu$ gives the geodesic equation in the presence of the dilaton field, namely (in the gauge $e(p) = \text{const.}$)

$$
\frac{D^2(g = \phi^2 \hat{g}) x^\lambda}{dp^2} = \frac{D^2(\hat{g}) x^\lambda}{dp^2} + 2 \frac{\partial_\mu \phi}{\phi} \frac{dx^\mu}{dp} \frac{dx^\lambda}{dp} - \frac{\partial^\lambda \phi}{\phi} \frac{dx^\mu}{dp} \frac{dx_\mu}{dp} = 0,
\tag{35}
$$

where $D^2(\hat{g})$ is the covariant derivative with respect to the metric $\hat{g}_{\mu\nu}$.

However, when we contract Equation (35) with the velocity $dx_\lambda/dp$ and we use $d\hat{s}^2 = 0$ obtained in (34), we obtain the following on-shell condition,

$$
\frac{dx_\lambda}{dp} \frac{D^2(\hat{g}) x^\lambda}{dp^2} = 0.
\tag{36}
$$

Therefore, the covariant derivative $\frac{D^2(\hat{g}) x^\lambda}{dp^2}$ must be proportional to the velocity, namely

$$
\frac{D^2(\hat{g}) x^\lambda}{dp^2} = f \frac{dx^\lambda}{dp} \quad (f = \text{const.})
\tag{37}
$$

because the velocity is null on the light cone. Under a reparametrization of the world line $q = q(p)$ Equation (37) becomes

$$\frac{d^2 x^\lambda}{dq^2} + \hat{\Gamma}^\lambda_{\mu\nu} \frac{dx^\mu}{dp} \frac{dx^\nu}{dp} = \frac{dx^\lambda}{dp} \left(\frac{dp}{dq}\right) \left(f \frac{dq}{dp} - \frac{d^2 q}{dp^2}\right). \tag{38}$$

Choosing the dependence of $q$ on $p$ such us to make vanish the right-hand side of (38), we end up with the geodesic equation in the affine parametrization. Hence, we can redefine $q \to \lambda$ and, finally, we obtain the affinely parametrized geodesic equation for photons in the metric $\hat{g}_{\mu\nu}$,

$$\frac{D^2(\hat{g}) x^\lambda}{d\lambda^2} = 0. \tag{39}$$

We can now investigate the conservation laws based on the symmetries of the metric $\hat{g}_{\mu\nu}$. Let us consider the following scalar,

$$\hat{\alpha} = \hat{g}_{\mu\nu} v^\mu \frac{dx^\nu}{d\lambda} = \hat{g}_{\mu\nu} v^\mu u^\nu. \tag{40}$$

where $v^\mu$ is a general vector and $u^\nu$ the four velocities. Taking the derivative of (40) with respect to $\lambda$ and using the geodesic Equation (39), we obtain:

$$\frac{d}{d\lambda} \hat{\alpha} = \frac{1}{2} v^\mu \partial_\mu \hat{g}_{\rho\nu} \frac{dx^\rho}{d\lambda} \frac{dx^\nu}{d\lambda} + \hat{g}_{\mu\nu} \partial_\rho v^\mu \frac{dx^\nu}{d\lambda} \frac{dx^\rho}{d\lambda} = \frac{1}{2} [\mathcal{L}_v \hat{g}]_{\rho\nu} \frac{dx^\rho}{d\lambda} \frac{dx^\nu}{d\lambda}, \tag{41}$$

where $[\mathcal{L}_v \hat{g}]$ is the Lie derivative of $\hat{g}_{\mu\nu}$ by a vector field $v^\mu$. Thus, if $v^\mu$ is a Killing vector field, namely $[\mathcal{L}_v \hat{g}] = 0$, $\hat{\alpha}$ is conserved:

$$\frac{d}{d\lambda} \left[\hat{g}_{\mu\nu} v^\mu \frac{dx^\nu}{d\lambda}\right] = 0. \tag{42}$$

The metric (7) is time-independent and spherically symmetric (in particular it is invariant under $t \to t + \delta t$ and $\varphi \to \varphi + \delta\varphi$). Therefore, we have the following Killing vectors associated with the above symmetries

$$\xi^\alpha = (1, 0, 0, 0), \quad \eta^\alpha = (0, 0, 0, 1). \tag{43}$$

Since the metric is independent of the $t$- and $\varphi$-coordinates, according to (40) we can construct the following conserved quantities

$$e = -\xi^\alpha u^\beta \hat{g}_{\alpha\beta} = -\hat{g}_{t\beta} u^\beta = -\hat{g}_{tt} u^t = Q^2(x) \left(1 - \frac{2M}{x}\right) \frac{dt}{d\lambda} = Q^2(x) \left(1 - \frac{2M}{x}\right) \dot{t}, \tag{44}$$

$$\ell = \eta^\alpha u^\beta \hat{g}_{\alpha\beta} = \hat{g}_{\phi\beta} u^\beta = \hat{g}_{\phi\phi} u^\phi = Q^2(x) x^2 \sin^2\theta \, \dot{\varphi}, \tag{45}$$

where the null vector

$$u^\alpha = \frac{dx^\alpha}{d\lambda} \tag{46}$$

satisfies

$$u \cdot u = \hat{g}_{\alpha\beta} \frac{dx^\alpha}{d\lambda} \frac{dx^\beta}{d\lambda} = 0, \tag{47}$$

as a consequence of (34).

From (47) in the equatorial plane (i.e., $\theta = \pi/2$), we obtain the following equation

$$-\left(1 - \frac{2GM}{x}\right)\dot{t}^2 + \frac{\dot{x}^2}{\left(1 - \frac{2GM}{x}\right)} + x^2\,\dot{\varphi}^2 = 0\,. \tag{48}$$

Note that the rescaling of the metric cancels out in the above Equation (48) for null geodesics, but $Q^2(x)$ will appear again when the conserved quantities (44) and (45) are taken into account. Let us solve (44) for $\dot{t}$ and (45) for $\dot{\varphi}$ and, afterward, replace the results in (48). The outcome is:

$$-\frac{e^2}{Q(x)^4\left(1 - \frac{2GM}{x}\right)} + \frac{\dot{x}^2}{1 - \frac{2GM}{x}} + \frac{\ell^2}{Q(x)^4 x^2} = 0\,. \tag{49}$$

Let us focus on the radial geodesics (i.e., $\ell = 0$), which will be sufficient to verify the geodesic completeness. Equation (49) simplifies to:

$$Q^2(x)|\dot{x}| = e\,. \tag{50}$$

The above first-order differential equation can be easily integrated for a photon traveling toward the boundary $x = 2/\gamma^*$, namely for $\dot{x} > 0$. The result of the integration is:

$$x(\lambda) = \frac{4\lambda - 2\gamma^*\lambda x_0 + 4x_0}{2\gamma^*\lambda - \gamma^{*2}\lambda - x_0 + 4}\,, \tag{51}$$

where $x_0$ is the initial position from which the photon is emitted, and

$$\lim_{\lambda \to +\infty} x(\lambda) = \frac{2}{\gamma^*}\,. \tag{52}$$

It turns out that photons cannot reach $x = 2/\gamma^*$ for any finite value of the affine parameter $\lambda$.

In the coordinate $r$, the geodesic Equation (50) turns into:

$$Q(x[r])^4\left(\frac{\partial x[r(\tau)]}{\partial r}\right)^2 \dot{r}^2 = e^2\,, \tag{53}$$

then we have

$$|\dot{r}| = e\,. \tag{54}$$

It is clear to see that a massless particle can reach $r = +\infty$ only for $\lambda = \infty$. The above Equation (54) has been derived in the Appendix A.1 also directly starting from the metric (10).

## 3. Uniqueness of the Solution

In the first part of this paper, the rescaling of the metric $Q(x)$ was chosen compatibly with the relation $g_{00} = -1/g_{11}$, as evident in the coordinate $r$. In this section, we would like to provide three fundamental reasons to support such a choice.

(i) The first one is related to the null energy condition, which asserts that $p + \rho \geqslant 0$ [28]. Indeed, to preserve the null energy condition we must impose $g_{00} = -1/g_{11}$.

(ii) The second one is related to the acceleration of the light in the Newtonian regime. Indeed, if the velocity of light has to remain constant in empty space surrounding a point-like mass, then photons should experience zero acceleration [29]. Using the last result in the previous subsection, namely $|\dot{r}| = e$ we obtain $\ddot{r} = 0$, which is true only if the relation

$g_{00} = -1/g_{11}$ for the components of the metric tensor is satisfied. Let us expand on this point. For a general spherically symmetric metric,

$$ds^2 = -A(r)dt^2 + B(r)dr^2 + r^2 d\Omega^2, \tag{55}$$

making use again of (44), namely

$$e = A(r)\dot{t}, \tag{56}$$

and $ds^2 = 0$, the radial geodesic equation reads

$$\dot{r}^2 = \frac{e^2}{A(r)B(r)}. \tag{57}$$

Then, we have

$$2\dot{r}\ddot{r} = \left(\frac{e^2}{A(r)B(r)}\right)' \dot{r}, \tag{58}$$

where $'$ means derivative respect to $r$. Finally,

$$\ddot{r} = \left(\frac{e^2}{2A(r)B(r)}\right)'. \tag{59}$$

Therefore, to not experience acceleration in the radial coordinate we must have: $A(r)B(r) = \text{const}$. Notice that here the radial coordinate is not the physical radial distance because the spacetime is not asymptotically flat. However, according to the Taylor expansion of (83) in the Newtonian intermedium regime $\ell_r \approx r$ the acceleration above vanishes.

(iii) Last but not least, we should consider the impact of the large distance modification of the Schwarzschild metric on the homogeneity and isotropy of the Universe.

Let us start by considering the following coordinate transformation from the radial coordinate $r$ to $\rho$,

$$\rho = \frac{4r}{2(1 + \alpha r + \beta r^2)^{1/2} + 2 + \alpha r}, \tag{60}$$

$$\tau = \int dt R(t), \tag{61}$$

in the following general not asymptotically flat metric,

$$d\hat{s}^{*2} = -\left(1 + \alpha r + \beta r^2\right)dt^2 + \frac{dr^2}{(1 + \alpha r + \beta r^2)} + r^2 d\Omega^{(2)}. \tag{62}$$

The above metric (62) in the new coordinates reads:

$$d\hat{s}^{*2} = \frac{1}{R^2(\tau)}\left[\frac{1 - \frac{\alpha^2 \rho^2}{16} + \frac{\beta\rho^2}{4}}{\left(1 - \frac{\alpha\rho}{4}\right)^2 - \frac{\beta\rho^2}{4}}\right]^2 \left\{-d\tau^2 + \frac{R(\tau)^2}{\left[1 - \left(\frac{\alpha^2}{16} - \frac{\beta}{4}\right)\rho^2\right]^2}\left(d\rho^2 + \rho^2 d\Omega^{(2)}\right)\right\}, \tag{63}$$

where $R(\tau) := R(t(\tau))$.

Now, in a geometry that is both homogeneous and isotropic about all points, any observer can serve as the origin of the radial coordinate $\rho$; thus, in their own local rest frame, each observer is able to make the above general coordinate transformation using their own particular $\rho$. Moreover, in conformal gravity, we can make an overall rescaling of the metric to finally end up with a comoving Robertson-Walker (RW) spacetime written in spatially isotropic coordinates with spatial curvature $K = \beta - \alpha^2/4$,

$$d\hat{s}^{*2} = F(\tau, \rho) \left[ -d\tau^2 + \frac{R(\tau)^2}{(1 + K\rho^2/4)^2} \left( d\rho^2 + \rho^2 d\Omega^{(2)} \right) \right].$$ (64)

For the case of the metric (10), taking $r \gg 2GM$ and $GM\gamma^* \ll 1$,

$$d\hat{s}^{*2} \approx -\left( 1 + \gamma^* r + \frac{\gamma^{*2}}{4} r^2 \right) dt^2 + \frac{dr^2}{\left( 1 + \gamma^* r + \frac{\gamma^{*2}}{4} r^2 \right)} + r^2 d\Omega^{(2)}.$$ (65)

Therefore, we can identify the constants $\alpha = \gamma^*$ and $\beta = \gamma^{*2}/4$, and in the new coordinates $(\tau, \rho)$ the metric (65) takes the following RW form,

$$d\hat{s}^{*2} = \frac{1}{R^2(\tau)} \frac{1}{\left( 1 - \frac{\gamma^*}{2} \rho \right)^2} \left[ -d\tau^2 + R(\tau)^2 \left( d\rho^2 + \rho^2 d\Omega^{(2)} \right) \right],$$ (66)

which coincides with the metric (7) for $x \gg 2GM$ upon reintroducing the time coordinate $t$ defined in (61).

Therefore, the metric proposed in this paper is the *only one* that does not affect the homogeneity of the Universe at large scales. Finally, we notice that the metric (65) is asymptotically (for large $r$) anti-de Sitter, whose stability is guaranteed from the fact that it comes from a rescaling of the Schwarzschild metric, which is known to be stable.

## 4. The Orbital Velocity

In this section, we compute the orbital velocity of a conformally coupled probe particle on the equatorial plane in the geometry (10) and (7), respectively, assuming zero radial velocity. For completeness let us remember here the action for a conformally coupled particle (19),

$$S_{\text{cp}} = -\int \sqrt{-f^2\phi^2 \hat{g}_{\mu\nu} dx^\mu dx^\nu} = -\int \sqrt{-f^2\phi^2 \hat{g}_{\mu\nu} \frac{dx^\mu}{d\lambda} \frac{dx^\nu}{d\lambda}} \, d\lambda,$$ (67)

from which the Lagrangian reads:

$$L_{\text{cp}} = -\sqrt{-f^2\phi^2 \hat{g}_{\mu\nu} \dot{x}^\mu \dot{x}^\nu}.$$ (68)

Since both the metrics (10) and (7) are invariant whether we make the replacements $t \to t + \text{const.}$ and $\varphi \to \varphi + \text{const.}$. Therefore, from the Lagrangian (68) we obtain the following conserved quantities (for $\theta = \pi/2$),

$$\frac{\partial L_{\text{cp}}}{\partial \dot{t}} = -\frac{f^2\phi^2 \hat{g}_{tt} \dot{t}}{L_{\text{cp}}} = -E, \qquad \frac{\partial L_{\text{cp}}}{\partial \dot{\varphi}} = -\frac{f^2\phi^2 \hat{g}_{\varphi\varphi} \dot{\varphi}}{L_{\text{cp}}} = \ell.$$ (69)

In the proper time gauge $\lambda \equiv \tau$, $d\hat{s}^2/d\lambda^2 = -1$ and $L_{\text{cp}} = -f\phi$. Hence, from (69),

$$\dot{t} = \frac{E}{f\phi \hat{g}_{tt}}, \qquad \dot{\varphi} = -\frac{\ell}{f\phi \hat{g}_{\varphi\varphi}}.$$ (70)

### 4.1. The Orbital Velocity in the Metric (10)

Let us in this section focus on the metric (10). Again in the proper time gauge and for $\theta = \pi/2$, the geodesics equation reads

$$\hat{g}_{tt} \dot{t}^2 + \hat{g}_{rr} \dot{r}^2 + \hat{g}_{\varphi\varphi} \dot{\varphi}^2 = -1,$$ (71)

and replacing (70) in (71), we obtain:

$$\frac{E^2}{f^2\,\phi^2\,\hat{g}_{tt}} + \hat{g}_{rr}\dot{r}^2 + \frac{\ell^2}{f^2\,\phi^2\,\hat{g}_{\varphi\varphi}} = -1\,. \tag{72}$$

Since we are interested in the orbital motion we can take $\dot{r} = 0$ and we end up with the following constraint equation,

$$-\frac{E^2}{f^2\,Q^{-2}(r)\kappa_4^{-2}\,Q^2(r)\left(1 - \frac{2GMQ(r)}{r}\right)} + \frac{\ell^2}{f^2\,Q^{-2}(r)\kappa_4^{-2}\,r^2} = -1\,. \tag{73}$$

In order to extract a simple relation for the ratio between $\ell^2$ and $E^2$, we take the derivative of Equation (73) respect to $r$,

$$E^2\,\frac{\frac{d}{dr}\left(1 - \frac{2GMQ(r)}{r}\right)}{\left(1 - \frac{2GMQ(r)}{r}\right)^2} + \ell^2\,\frac{d}{dr}\left(\frac{Q^2(r)}{r^2}\right) = 0\,. \tag{74}$$

Then, we obtain

$$\implies \quad \frac{\ell^2}{E^2} = \frac{2GMr^3}{(2 + r\gamma^*)(r - GM(2 + r\gamma^*))^2}\,. \tag{75}$$

The physical velocity on the equatorial plane and along the $\varphi$-direction reads:

$$v = \frac{\sqrt{\hat{g}_{\varphi\varphi}}}{\sqrt{-\hat{g}_{tt}}}\frac{d\varphi}{dt} = \frac{\sqrt{\hat{g}_{\varphi\varphi}}}{\sqrt{-\hat{g}_{tt}}}\frac{\dot{\varphi}}{\dot{t}}\,, \tag{76}$$

where the dot stays for the derivative with respect to the proper time $\tau$. Replacing $\dot{t}$ and $\dot{\varphi}$ in (70) into (76) we obtain:

$$v^2 = -\frac{\hat{g}_{tt}}{\hat{g}_{\varphi\varphi}}\frac{\ell^2}{E^2}\,, \tag{77}$$

where we finally replace (75),

$$v^2 = \frac{GM(2 + \gamma^*r)}{2(r - GM(2 + \gamma^*r))} = \frac{GMQ(r)}{r - 2GMQ(r)}\,. \tag{78}$$

In the limit of $r \gg 2GM$, namely far from the Schwarzschild radius, the velocity turns into:

$$v^2 = \frac{GMQ(r)}{r(1 - GM\gamma^*)}\,, \tag{79}$$

and if we also assume $GM\gamma^* \ll 1$,

$$v^2 = \frac{GMQ(r)}{r} = \frac{GM}{r} + \frac{GM\gamma^*}{2} \tag{80}$$

which asymptotically approaches the constant value:

$$v^2 \to v_\infty^2 = \frac{GM\gamma^*}{2}\,. \tag{81}$$

Let us now express the velocity in terms of the physical length $\ell_r$ in place of the radial coordinate $r$. What we need is the physical radial length, namely

$$\begin{aligned}
\ell_r &= \int \sqrt{\hat{g}_{rr}}\, dr + \text{const} = \int \frac{dr}{Q(r)\sqrt{1 - \dfrac{2GMQ(r)}{r}}} + \text{const} \\
&\approx \int \frac{dr}{\sqrt{1 - GM\gamma^*}\,\left(1 + \dfrac{\gamma^*}{2} r\right)} + \text{const} = \frac{2\log(2 + \gamma^* r)}{\gamma^*\sqrt{1 - GM\gamma^*}} + \text{const}.
\end{aligned} \tag{82}$$

where in the last two steps we have integrated for $r \gg 2GM$. Finally, we fix the integration constant imposing that $\ell_r(r = 0) = 0$,

$$\ell_r = \frac{2\log\left(1 + \frac{\gamma^*}{2} r\right)}{\gamma^*\sqrt{1 - GM\gamma^*}}. \tag{83}$$

Notice that in the intermedium Newtonian regime, namely $r \ll 2/\gamma^*$, and for $GM\gamma^* \ll 1$, $\ell_r \approx r$. The inverse relation $r(\ell_r)$ reads:

$$r(\ell_r) = \frac{2\left(e^{\frac{1}{2}\gamma^*\ell_r\sqrt{1-GM\gamma^*}} - 1\right)}{\gamma^*} \approx \frac{2\left(e^{\frac{1}{2}\gamma^*\ell_r} - 1\right)}{\gamma^*}, \tag{84}$$

where the last approximation comes again from $GM\gamma^* \ll 1$ (notice that also $r(\ell_r = 0) = 0$).

Replacing (84) in (79), we obtain the physical velocity square, namely

$$v^2(\ell_r) = \frac{GM\gamma^*}{4} \frac{\left[1 + \coth\left(\frac{1}{4}\ell_r\gamma^*\sqrt{1 - GM\gamma^*}\right)\right]}{1 - GM\gamma^*}, \tag{85}$$

which further simplifies for $GM\gamma^* \ll 1$,

$$\boxed{v^2(\ell_r) = \frac{GM\gamma^*}{4}\left[1 + \coth\left(\frac{\ell_r\gamma^*}{4}\right)\right].} \tag{86}$$

The above astonishing simple analytic result correctly interpolates between Newtonian velocity and the asymptotic constant value (81). It deserves to be noticed that for small $\gamma^*$, namely $\ell_c \gg r_g = 2GM$ ($r_g$ is the Schwarzschild radius), the exact result (85) and the velocity (80) are extremely close to each other. Therefore, the following replacement is a good approximation of (80),

$$v^2(\ell_r) = \frac{GM}{\ell_r} + \frac{GM\gamma^*}{2}. \tag{87}$$

### 4.2. The Orbital Velocity in the Metric (7)

In this section, we compute again the velocity square, but now for the metric (7). This computation not only will provide a further check of our result (86), but also will make more explicit the crucial role of the asymptotic singularity in $x = 2/\gamma^* = \ell_c$.

According to the previous section ($\dot{t}$ and $\dot{\varphi}$) (70), the velocity (76), and the velocity square (77) are general and independent of the metric. However, the ratio $\ell^2/E^2$ does depend on the metric. Indeed, the proper time gauge for the metric (7) reads:

$$\frac{d\hat{s}^2}{d\lambda^2} = -1 \implies \hat{g}_{tt}\dot{t}^2 + \hat{g}_{xx}\dot{x}^2 + \hat{g}_{\varphi\varphi}\dot{\varphi}^2 = -1 \tag{88}$$

which, for $x = $const. and replacing the metric (7) within, turns into:

$$\hat{g}_{tt}\left(\frac{E}{f\,\phi\,\hat{g}_{tt}}\right)^2 + \hat{g}_{\varphi\varphi}\left(-\frac{\ell}{f\,\phi\,\hat{g}_{\varphi\varphi}}\right)^2 = -1 \quad \Longrightarrow \quad \frac{E^2}{f^2\,\phi^2\,\hat{g}_{tt}} + \frac{\ell^2}{f^2\,\phi^2\,\hat{g}_{\varphi\varphi}} = -1$$

$$\Longrightarrow \quad \frac{E^2}{f^2\,Q^{-2}(x)\,Q^2(x)\left(1 - \dfrac{2GM}{x}\right)} + \frac{\ell^2}{f^2\,Q^{-2}(x)\,Q^2(x)x^2} = -1 \tag{89}$$

$$\Longrightarrow \quad \frac{E^2}{f^2\left(1 - \dfrac{2GM}{x}\right)} + \frac{\ell^2}{f^2\,x^2} = -1\,,$$

which is independent of the rescaling $Q(x)$. Taking the derivative of (89) respect to radial coordinate $x$, we find:

$$\frac{\ell^2}{E^2} = -\frac{g'_{tt}}{g'_{\varphi\varphi}}\frac{g^2_{\varphi\varphi}}{g^2_{tt}}\,. \tag{90}$$

where we defined:

$$g_{tt} = -\left(1 - \frac{2GM}{x}\right), \quad g_{\varphi\varphi} = x^2\,. \tag{91}$$

The one above is not just a definition, but the Schwarzschild metric before introducing the rescaling $Q(x)$. Substitution of (90) in the velocity square (77) and making use of (91) together imply:

$$v^2(x) = \frac{g'_{tt}}{g'_{\varphi\varphi}}\frac{g_{\varphi\varphi}}{g_{tt}} = \frac{GM}{x - 2GM} \approx \frac{GM}{x}\,, \tag{92}$$

where in the last equality we assumed $x \gg 2GM$.

The result just found for the velocity square may seem trivial and obvious, but it is actually rich in geometric meaning. Indeed, it is exactly the Newtonian result in the radial coordinate $x$. However, we must remember that the larger value for $x$ is $2/\gamma^*$ and, therefore, the minimum asymptotic value for the velocity square is $GM\gamma^*/2$ in perfect agreement with (81). This is clearly due solely to the singular structure of the conformal geometry in the unattainable asymptotic point $x = 2/\gamma^*$.

To complete the section we now express the velocity square in terms of the physical length $\ell_x$ that we set about calculating,

$$\begin{aligned}\ell_x &= \int \sqrt{g_{xx}}\,dx + \text{const.} = \int dx\,\frac{Q(x)}{\sqrt{1 - \dfrac{2GM}{x}}} + \text{const.} \approx \int Q(x)\,dx + \text{const.} \\ &= -\frac{2}{\gamma^*}\log\left(1 - \frac{\gamma^* x}{2}\right),\end{aligned} \tag{93}$$

where again we assumed $x \gg 2GM$ and we fixed the integration constant imposing $\ell_x(x = 0) = 0$. Notice that $\ell_x \to +\infty$ for $x \to 2/\gamma^*$.

It is straightforward to invert (93),

$$x(\ell_x) = \frac{2}{\gamma^*}\left(1 - e^{-\frac{\ell_x \gamma^*}{2}}\right). \tag{94}$$

Replacing the above expression in the velocity square (92) we find:

$$v^2(\ell_x) = \frac{GM}{x(\ell_x)} = \frac{GM\gamma^*}{4}\left[1 + \coth\left(\frac{\ell_x \gamma^*}{4}\right)\right], \tag{95}$$

which of course agrees with (86), which is also expressed in terms of the physical distance. Notice that $\ell_x \equiv \ell_r$ because there is only one physical observable distance in nature.

A further remark in comparison with the existing literature on conformal gravity is needed. In particular, we will focus on the solution proposed in [30] and the issue pointed out in [31].

To preserve the conformal symmetry, in this paper the matter has been coupled to gravity in a conformal invariant way. Hence, in the metric (7), which is conformally equivalent to the Schwarzschild solution, the motion of conformally coupled massive probe particles is the same as in the Schwarzschild metric. However, since the conformal rescaling $Q(x)$ (see again the metric (7)) is singular, the Schwarzschild coordinate $x$ has a cutoff at the value $x = 2/\gamma^* \equiv \ell_c$, which is actually the causal infinity because to reach such limit we need an infinite amount of proper time. Therefore, the velocity $v(x)$ (see (92) and the discussion in the text right after) has a lower positive bound $v(x = \ell_c) = GM\gamma^*/2 > 0$, which is consistent with asymptotically flat rotation curves. Therefore, contrary to [30], our model can explain the rotation curves on a purely geometrical ground and consistently with Hobson and Lasenby's analysis [31].

## 5. Newtonian Effective Theory and Gravitational Potential

To derive the effective gravitational potential, we start from the orbital velocity in terms of the physical distance. Indeed, in Newtonian physics, we only deal with physical lengths, and the Lagrangian simply reads:

$$\mathcal{L}_N = \frac{1}{2}m\left(\frac{d\vec{r}}{dt}\right)^2 - m\Phi(|\vec{r}|) = \frac{1}{2}m\left(\dot{\ell}_r^2 + \ell_r^2\dot{\varphi}^2\right) - m\Phi(\ell_r), \tag{96}$$

where $|\vec{r}| = \ell_r$, $m$ is the mass of a probe particle, and we assumed to be on the equatorial plane $\theta = \pi/2$. From the Lagrangian above the EoM, assuming $\dot{\ell}_r = 0$, is:

$$\ell_r\dot{\varphi}^2 = \frac{v^2(\ell_r)}{\ell_r} = \frac{\partial\Phi(\ell_r)}{\partial\ell_r} = -E_r(\ell_r) \implies v^2(\ell_r) = -\ell_r E_r(\ell_r), \tag{97}$$

where for future reference we also defined the gravitational field $\vec{E} = -\vec{\nabla}\Phi$.

Therefore, the effective potential can be obtained simply by integrating (86) or (95),

$$\Phi(\ell_r) = \int d\ell_r' \frac{v^2(\ell_r')}{\ell_r'} + \text{const.}. \tag{98}$$

However, the velocity in (98) can be very well approximated making use of (87), and the integral (98) can be easily computed to give the following result,

$$\Phi(\ell_r) \approx -\frac{GM}{\ell_r} + \frac{GM\gamma^*}{2}\log(\ell_r) + \text{const.}. \tag{99}$$

Now we have to consider the contribution of all the stars in a galaxy gravitationally acting on a probe star. This consists of integrating the potential in cylindrical coordinates after having introduced the following three vectors: $\vec{R}$, which points from the center of the galaxy to the probe star, $\vec{R}'$ from the center of the galaxy to one of its stars, and $\vec{r}$ pointing from a star in the galaxy to the probe star. Therefore, we have: $\vec{r} = \vec{R} - \vec{R}'$ and the contribution to the potential due to any star in the galaxy is:

$$\Phi(|\vec{R} - \vec{R}'|) = -\frac{GM}{|\vec{R} - \vec{R}'|} + \frac{GM\gamma^*}{2}\log(|\vec{R} - \vec{R}'|) + \text{const.} \equiv \Phi_0 + \Phi^{\log}. \tag{100}$$

Notice that we replaced $\ell_r$ with $|\vec{R} - \vec{R}'|$ because the Newtonian effective theory is defined in flat spacetime.

Let us now consider a thin disk galaxy model with an exponential distribution of matter that decays at large distances. We assume that the mass of each star is $M_\odot$ (solar

mass) and the distribution of stars is described as follows (in cylindrical coordinates: $R, \varphi, z$):

$$\rho(R', z') = \Sigma_0 \, e^{-\frac{R'}{R_0}} \, \delta(z') \,, \quad [\rho] = L^{-3} \,, \tag{101}$$

where $z$ is the coordinate orthogonal to the galaxy plane. Moreover, $R_0$ is the radius of the galaxy and $\Sigma_0$ is related to the number of stars of mass comparable to the solar mass in the galaxy, i.e.,

$$
\begin{aligned}
N^* &= \int_0^{+\infty} dR' \, R' \int_0^{2\pi} d\varphi' \int_{-\infty}^{+\infty} dz' \, \rho(R', z') \\
&= \int_0^{+\infty} dR' \, R' \int_0^{2\pi} d\varphi' \int_{-\infty}^{+\infty} dz' \Sigma_0 \, e^{-\frac{R'}{R_0}} \, \delta(z') = 2\pi \Sigma_0 R_0^2 \\
&\implies \quad \Sigma_0 = \frac{N^*}{2\pi R_0^2} \,.
\end{aligned}
\tag{102}
$$

In order to obtain the total contribution to the gravitational potential we have to integrate over all the stars in the galaxy each of them of solar mass $M_\odot$, namely:

$$\Phi_{\mathrm{T}}(R, z) = \int_0^{+\infty} dR' \, R' \int_0^{2\pi} d\varphi' \int_{-\infty}^{+\infty} dz' \, \rho(R', z') \, \Phi(R, R', z, z') \,,$$

$$
\begin{aligned}
\Phi(R, R', z, z') = \quad &-\frac{GM_\odot}{\left[R^2 + R'^2 - 2R\,R' \cos\varphi' + (z - z')^2\right]^{\frac{1}{2}}} \\
&+ \frac{GM_\odot \gamma^*}{2} \log\left\{ \frac{\left[R^2 + R'^2 - 2R\,R' \cos\varphi' + (z - z')^2\right]^{\frac{1}{2}}}{\ell} \right\} \,,
\end{aligned}
\tag{103}
$$

where $R$ is the distance of the probe star from the galactic center in cylindrical coordinates and $R_0$ is the characteristic scale of the galaxy. Since $\Phi(R, R', z, z')$ consists of two parts, we will integrate the two contributions of the potential separately obtaining the two corresponding contributions to the velocity square. Finally, $\ell$ is the scale coming from the integration constant that cannot be zero since the potential grows with the distance. However, we do not have to worry about such scale because it will disappear in the orbital velocity that is related to the force and not the potential.

For the Newtonian potential contribution to (100), namely $\Phi_0 = -GM/|\vec{R} - \vec{R'}|$, and assuming the density profile (101), the rotation velocity square of a probe star was computed in [32] and the result is:

$$v_0^2 = \frac{GN^* M_\odot R^2}{2R_0^3}\left[ I_0\left(\frac{R}{2R_0}\right) K_0\left(\frac{R}{2R_0}\right) - I_1\left(\frac{R}{2R_0}\right) K_1\left(\frac{R}{2R_0}\right) \right] \,, \tag{104}$$

where $I_0, I_1$ are the modified Bessel functions of the first kind and $K_0, K_1$ are the modified Bessel functions of the second kind. In (104) $M = N^* M_\odot$ is the mass of all stars in the galaxy.

To compute the logarithmic contribution to the potential in (100), namely

$$\Phi^{\log} = \frac{GM\gamma^*}{2} \log \frac{|\vec{R} - \vec{R'}|}{\ell} \,, \tag{105}$$

we can use the Gaussian theorem

$$\mathrm{div}\vec{E} = -4\pi G\rho \,, \tag{106}$$

to sum up all the stars in the galactic disk. Notice that we can assume the sources of the logarithmic potential to be wires because of the cylindrical symmetry of the galaxy.

Due to the above logarithmic correction (105), the gravitational field in cylindrical coordinates (we here fix the origin in $\vec{R}' = 0$) is attractive and reads:

$$E_R = -\frac{\partial \Phi^{\log}(R)}{\partial R} = -\frac{GM\gamma^*}{2R}. \tag{107}$$

Integrating (106) on a three-dimensional volume $V$ with boundary $\partial V$ in Cylindrical coordinates, we can infer the energy density $\rho_s(\vec{x}) = \rho_0 \delta(x)\delta(y)$ of a single wire-like source, namely

$$\int_V \operatorname{div}\vec{E}\, dv = \operatorname{Flux}\Big|_{\partial V}(E) = -4\pi G \int_V dv \rho_s(\vec{x})\,,$$
$$2\pi R \Delta z\, E_R = -4\pi G \rho_0 \Delta z\,. \tag{108}$$

Replacing (107) in (108) we finally find $\rho_0$,

$$\rho_0 = -\frac{R\, E_R}{2} = \frac{M\gamma^*}{4}\,, \tag{109}$$

and the potential can be recast in the following form in terms of the energy density,

$$\Phi^{\log} = 2G\rho_0 \, \log\frac{|\vec{R} - \vec{R}'|}{\ell}\,. \tag{110}$$

If the gravitational sources and the probe star are all located in the same plane (we here assume the galactic disk to be in $z = 0$ in cylindrical coordinates), then $\Phi^{\log}$ is analogous to the Newtonian potential of $N^*$ massive infinite wires each with uniform density $\rho_0$ and generating a logarithmic gravitational potential.

Assuming the principle of the linearity of the gravitational force and then of the gravitational potential, we can now apply again Gauss' theorem to all the stars in the galaxy that are described by the energy density profile in cylindrical coordinates:

$$\rho_s(\vec{x}) = \rho_0 \delta(x)\delta(y) \quad\longrightarrow\quad \rho_{N^*}(R) = \rho_0 \Sigma_0\, e^{-\frac{R}{R_0}} = \frac{M_\odot \gamma^*}{4}\Sigma_0\, e^{-\frac{R}{R_0}}\,, \quad [\rho_{N^*}] = ML^{-3}, \tag{111}$$

where we assumed any star to have mass $M_\odot$. Notice that (111) is an energy density while (101) is a density distribution.

Finally, the Gaussian theorem making use of the above energy density (111) gives:

$$-2\pi R \Delta z\, E_R^{\mathrm{T}}(R) = 4\pi G \int_0^R \int_0^{2\pi} R' dR' d\varphi\, \rho_{N^*}(R') \int_0^{\Delta z} dz$$
$$\implies\quad E_R^{\mathrm{T}}(R) = -\frac{4\pi G}{R}\int_0^R \rho_{N^*}(R')R'dR'\,. \tag{112}$$

Using (97) and upon integration of (112), the contribution to the rotation velocity square (97) due to the logarithmic term in the potential reads,

$$v_{\log}^2 = -E_R^{\mathrm{T}}(R)R = \pi G M_\odot \gamma^* \Sigma_0 \int_0^R e^{-\frac{R'}{R_0}}R'dR' = \frac{GM_\odot \gamma^*}{2}2\pi\Sigma_0 R_0^2\left[1 - \left(1 + \frac{R}{R_0}\right)e^{-\frac{R}{R_0}}\right]$$
$$= \frac{GN^* M_\odot \gamma^*}{2}\left[1 - \left(1 + \frac{R}{R_0}\right)e^{-\frac{R}{R_0}}\right]. \tag{113}$$

Finally, taking the sum of (104) and (113) the total contribution to the velocity square reads:

$$\boxed{\begin{aligned} v^2(R) = {}&\frac{GN^* M_\odot R^2}{2R_0^3}\left[I_0\left(\frac{R}{2R_0}\right)K_0\left(\frac{R}{2R_0}\right) - I_1\left(\frac{R}{2R_0}\right)K_1\left(\frac{R}{2R_0}\right)\right] \\ &+ \frac{GN^* M_\odot \gamma^*}{2}\left[1 - \left(1 + \frac{R}{R_0}\right)e^{-\frac{R}{R_0}}\right], \end{aligned}} \tag{114}$$

which is constant for large $R$, namely

$$v^2(R) \quad \rightarrow \quad \frac{GN^*M_\odot\gamma^*}{2} \quad \text{for} \quad R \rightarrow +\infty. \tag{115}$$

We end this section with a short review of an interesting two-dimensional dilation gravity model [33,34] (generalizations are provided in [35]) that provides similar modifications to the gravitational potential. In such a theory the spacetime is asymptotically described by Rindler's metric and the gravitational potential grows linearly at large distances. The model studied in [34] is consistent with the Pioneer's anomaly and perhaps it improves the galactic rotation curves, but fails at the earth's distance. It deserves to be noticed that contrary to Einstein's conformal gravity, the scalar-tensor theory in [35] propagates another scalar degree of freedom because it is not conformal invariant. Moreover, in our model, we can reproduce the rotation curves for galaxies and galaxy groups by fitting only one parameter $\gamma^*$.

## 6. The Tully–Fisher Relation

As we have said several times, in conformal gravity, we are free to rescale the metric by an overall factor that will depend on at least one undetermined length scale. In our model, the length scale is $\ell_c = 2/\gamma^*$, which turns out to be of the same order of magnitude as the galaxy (see next section). However, if we focus our attention on a single star in the galaxy we can with equal naturalness fix $\ell_c$ to be comparable with either the Schwarzschild radius of the star or the galaxy extension. Indeed, these two are the characteristic scales of the system. On the other hand, if we were dealing with a single star in an empty universe, it would be natural to select $\ell_c$ proportional to the Schwarzschild radius of the star. Therefore, conceptually there is nothing wrong in selecting the free scale to be proportional to the galaxy extension, and actually, it seems the natural choice whether we are interested in the global properties of the galaxies. Furthermore, in conformal gravity, we have an extra scalar field, the dilaton, that does not propagate (the perturbation can always be fixed to zero by the mean of conformal symmetry), but satisfies its equation of motion whose solutions show up extra scales simply because of dimensional reasons and in accordance with the Mach's mechanical view of the Universe. In other words, the dilaton is responsible for the gravitational interaction from small to large distances through the presence of pole-like singularities, which are weighted by dimensional parameters, in the solution of its equation of motion.

The arguments above have an observational counterpart in the Tully–Fisher relation that relates the asymptotic velocity of a probe star to Newton's constant, the mass of the galaxy, and Milgrom's parameter $a_0$, namely

$$v^4 = a_0 GM, \quad [a_0] = L\,T^{-2}, \quad [G] = L^3\,M^{-1}\,T^{-2}, \tag{116}$$

where $M = N^*M_\odot + M_{\text{HI}}$, $M_{\text{HI}}$ is the mass of the Helium gas (see next section for more details). Comparing the letter expression (116) with (81) we finally obtain:

$$\gamma^* = \sqrt{\frac{4a_0}{GM}}, \quad [\gamma^*] = L^{-1}, \tag{117}$$

which depends on the mass of the galaxy while we assume $a_0$ to be a universal constant.

For the value of $a_0$ obtained by fitting the galactic rotation curves with the MOND theory [36], namely $a_0 = 1.2 \times 10^{-10}\,\text{m}\,\text{s}^{-2}$, and for a galaxy made of $10^{12}$ solar mass stars we obtain:

$$\gamma^* \approx 10^{-21}\text{m}^{-1} \quad \Longrightarrow \quad \ell_c \approx 10^{21}\text{m}. \tag{118}$$

In conformal gravity, $\gamma^*$ is one of the two free parameters to be obtained by fitting the observational data and assuming dependence on the mass of the entire galaxy like in (117).

In the next section, we will obtain a universal value or $a_0$ from our model fitting 175 galaxies.

*The Solar System's Geometry*

For the solar system $1/\gamma^* \approx 10^{15}$ m, and, hence, the product $\gamma^* r1$ inside the solar system is very small, namely $\gamma^* r \ll 1$. Indeed, taking as the radius of the solar system the average distance between the sun and Pluto, namely $6 \times 10^{12}$ m, we obtain $\gamma^* r \sim 10^{-3} \ll 1$. On the other hand, if we consider the larger radius of the Oort Cloud, which is about $7.5 \times 10^{15}$ m, then $\gamma^* r \sim 1$. However, the above constraints in the solar system cannot be taken seriously because when we consider the solar system we cannot ignore the rest of the galaxy of which the sun is an integral part. Therefore, the value of $\gamma^*$ has to be fixed according to Tulley–Fisher which takes into account a large number of stars. In other words, the value of $\gamma^*$ in our universe is uniquely fixed on the base of observations at the galactic scale of which the sun is inseparably part. The latter statement is strictly in line with the Macchian view of a holistic essence of the Universe at large scales. The value of the mass of the sun could be correctly replaced in $\gamma^*(M)$ in a universe only consisting of the solar system or in a universe in which the solar system is not part of any galaxy. However, such a prediction is not falsifiable in the Popperian sense.

To avoid a large value for the quantity $\gamma^* r$ for a small mass compact object, we could also choose $\gamma^*$ as a less trivial function of the mass. In such a way, a simple proposal that can surely address the problem is the following function,

$$\gamma^* = \sqrt{\frac{4a_0}{G(M + M_0)}}, \tag{119}$$

where $M_0$ is a mass much larger than the solar system mass, but much smaller than the galaxy's mass.

We reiterate that in our opinion the most convincing argument is not of an engineering nature like the one just provided, but of a holistic nature, namely we cannot consider the solar system or any compact object in the Universe as independent from everything else. The function $\gamma^*(M)$ has to be the outcome of a comparison of the model with all the data, at the galactic as well as the extragalactic scales (galaxy groups and galaxy clusters). On the other hand, on the cosmological scale, the homogeneity and isotropicity of the Universe forces the spacetime to the FRW's metric.

Finally, we remember that the rescaling of the metric does not affect either the light bending, the Mercury precession, or other observables that are conformal invariant.

## 7. Fitting of the Galactic Rotation Curves and Universality

To completely specify the velocity square (114), we need $N^*$ (the number of stars in the galaxy), $R_0$ (the effective scale of the galactic disk), and the free scale in our model, namely $\gamma^*$. Moreover, we have to consider the contribution to the velocity due to the gas Helium (HI). If we apply to the HI the disk model with an exponential profile, the contribution of HI to $v^2$ will be described by the same formula (114). Therefore, the total $v^2$ reads:

$$v_{\text{tot}}^2 = v^2(N^*, R_0, \gamma^*) + v^2(N_{\text{HI}}, R_{\text{H0}}, \gamma^*), \tag{120}$$

where $N_{\text{HI}} = M_{\text{HI}}/M_\odot$ represents the fraction of the total mass of the HI gas with respect to the solar mass and $R_{\text{H0}}$ is the effective radius of the HI gas' cloud.

In our analysis, we used the data from the SPARC database [37] that includes: the rotation curves data, which the reader can find in the plots in Appendix D, the total luminosity ratio $L/L_\odot$, and the disk radius $R_0$(kpc) for 175 galaxies (see Appendix C). The database includes also $M_{\text{HI}}$, while $R_{\text{H0}}$ will be determined shortly. Of course, the mass $M_\odot$ and the luminosity $L_\odot$ of the sun, and the luminosity of all the galaxies $L$ are known observed quantities. All these parameters are given in Appendix C.

The number of stars $N^*$ is related to the mass-to-luminosity ratio $M/L$, which is our second fitting parameter, the ration $M_\odot/L_\odot$, and the ration $L/L_\odot$, namely

$$N^* = \frac{M}{M_\odot} = \frac{\frac{M}{L}}{\frac{M_\odot}{L_\odot}}\frac{L}{L_\odot}, \tag{121}$$

in which $M_\odot/L_\odot$, and $L/L_\odot$ are known and given in the table in Appendix C. Therefore, fitting $M/L$ is equivalent to the fitting of $N^*$. Since we assume that there is no dark matter, the fitting results of $M/L$ should be close to 1 rather than over 10 like in Newtonian dynamics.

In the database [37] we can also find the mass $M_{\mathrm{HI}}$. However, to also include the amount of primordial Helium, we have to multiply HI times the factor 1.4. Therefore, the total amount of Helium is:

$$M_{\mathrm{HI}}^{\mathrm{TOT}} = 1.4\,M_{\mathrm{HI}}. \tag{122}$$

In the SPARC database [37] one can find the radius $R_{\mathrm{H}}$ defined to be one for which the density of HI is equal to the value $M_\odot/\mathrm{pc}^2$. Therefore, we can infer the effective radius $R_{\mathrm{H0}}$ of the Helium gas using the exponential density profile (101) and (102),

$$\Sigma_{\mathrm{H0}}e^{-R_{\mathrm{H}}/R_{\mathrm{H0}}} = \frac{N_{\mathrm{HI}}}{2\pi R_{\mathrm{H0}}^2}e^{-R_{\mathrm{H}}/R_{\mathrm{H0}}} = \frac{1}{\mathrm{pc}^2} \quad \Longrightarrow \quad R_{\mathrm{H0}}, \tag{123}$$

where the parameters $N_{\mathrm{HI}}$, which can be identified with the dimensionless quantity $M_{\mathrm{HI}}$, are available in Appendix C. However, Equation (123) is ambiguous because it usually has two solutions. Moreover, for some galaxies, Equation (123) has no solutions, which implies that for these galaxies the measurements of $N_{\mathrm{HI}}$ and $R_{\mathrm{H}}$ are not accurate enough or the distribution of HI does not fit the disk model properly. Therefore, we choose $R_{\mathrm{H0}} = 4R_0$ as an effective radius of the HI disk consistently with other papers in the literature [1,32].

The results for the fitting parameters $M/L$ and $\gamma^*$ are given in Appendix C, while the fittings of the rotation curves are displayed in Appendix D.

The fitting results show that our model fits the rotation velocity data for most of the typical spiral galaxies (including S0, Sa, Sb, Sc, Sab, Sbc, and Scd type) and it fits very well some late spiral-type galaxies (Sd, Sdm, and Sm), in particular for the velocity data on the large scale ($R > 2R_0$).

As we expected, the fitting results for the mass-to-luminosity ratio (of luminous mass) are close to 1. Moreover, in the plots in Appendix D, we can see that the Newtonian contribution dominates the rotation velocity at a small scale ($R \lesssim 2R_0$), while the conformally modified geometry determines the value of the velocity square asymptotically. Our model (114) interpolates between the two regimes.

However, there are some galaxies to which our model cannot fit very well.

This is the case of the galaxies NGC3949, NGC3953, and NGC4051. However, for such galaxies, we have only a few data, and in particular, we lack data points at large radius. In this case, the fitting results for $\gamma^*$ is actually 0.

For some spiral galaxies, e.g., NGC2955, NGC5005, NGC6195, UGC2916, UGC3546, UGC5253, and UGC11914, the rotation velocity data tend to be flat at very small scales ($R \ll 2R_0$). Therefore, we think that the rotation curves cannot be consistent with the exponential profile for the matter density adopted.

For the irregular galaxies, Im (irregular Magellanic), BCD (irregular blue compact dwarf), and weak spiral types (Sm, Sd, and, Sdm), for instance: CamB, DDO161, F574-2, NGC2366, NGC3741, NGC4068, PGC51017, UGC2455, UGC4483, some fits are bad and usually the fitting results of the mass-to-luminosity ratio are anomalously small. However, this should be related to the irregular mass distribution of these galaxies that affect the irregular motion of matters.

Finally, having at our disposal the values of the fitting for $\gamma^*$ and $M/L$ ($L$ is an observed quantity) we can now extract the *universal* parameter $a_0$ using the Tully–Fisher relation (117). The total mass in (117) consists of two contributions, stars and Helium, namely

$$M = L \cdot \left( \frac{M}{L} \right) + 1.4 \, M_{\text{HI}} \,. \tag{124}$$

Let us consider the following generalization of Equation (117), namely

$$\gamma^* = \left( \frac{4a_0}{GM} \right)^k , \tag{125}$$

where the constant $k$ has to be determined by means of the fitting. Hence, taking the "log" of both sides we obtain:

$$\log \gamma^* = k(\log 4a_0 - \log GM) , \tag{126}$$

in which the fitting parameters are $a_0$ and $k$. The fitting results are shown in Figure 1 (notice that we removed the seven points for which $\gamma^* = 0$),

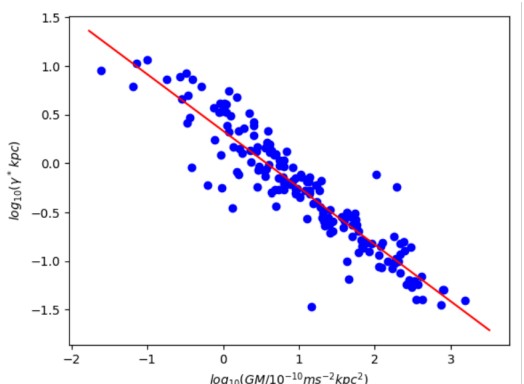

**Figure 1.** This plot shows the fitting of the relation between $\gamma^*$ and $M$. The fitting function is $y = k(b - x)$, where $y = \log((\gamma^*) \cdot \text{kpc})$, $x = \log(GM/10^{-10}\text{ms}^{-2}\text{kpc}^2)$ and $b = \log(4a_0/10^{-10}\text{ms}^{-2})$. The results for the two fitting parameters are: $k = 0.582$ and $b = 0.573$.

Where

$$\gamma^* \propto M^{-0.582}, \quad a_0 = 0.935 \times 10^{-10}\text{m/s}^2 = 9.35 \times 10^{-11}\text{m/s}^2 \,. \tag{127}$$

The $3\sigma$ confidence intervals of $k$ and $a_0$ are: $0.582 \pm 0.057$ and $(9.35 \pm 2.22) \times 10^{-11}$ m/s$^2$, respectively. Notice that according to (117) $k$ is compatible with $1/2$.

## 8. Conclusions

We provided a geometrical mechanism capable of overcoming the long-standing issue of galactic rotation curves without any kind of exotic dark matter. We are aware that dark matter is a proposal to remove multiple issues in cosmology and astrophysics while there is no need for it in the colliders' physics, but we found an extremely interesting outcome to this project from both the theoretical and observational sides. From the theoretical point of view, the simple scalar-tensor Einstein's theory of gravity provides a kind of non-modified gravitational theory ghost-free and free of other instabilities. Indeed, the presence of the dilaton field on one side allows for other vacua without introducing other propagating degrees of freedom, on the other side introduces unattainable spacetime singularities that drastically modify the asymptotic spacetime structure from the micro to the macro.

Specifically, the effective Newtonian gravitational force, to which the stars of the galaxy are subject, is obtained starting from a "unique" (the metric depends only on one

extra scale $\ell_c = 2/\gamma^*$, see Section 3) spacetime geometry (7) or (10) (in two different coordinate systems) for a single star and summing up all the stars in the galaxy. The effective potential has the expected asymptotic logarithmic behavior characteristic of the minimal confinement, and the velocity turns out to be constant (see formulas (80) or (87) and (81)) at a large distance from the galactic center in agreement with the Tully–Fisher relation.

In the force of the effective gravitational potential with logarithmic asymptotic behavior, we derived for a single source and integrated all the stars of the galaxy with exponential density profiles to end up with the total potential. Hence, we obtained the orbital velocity of a probe star in the gravitational field of all the other stars in the galaxy (see (114)). Afterward, we tested the theory with 175 galaxies, making a fit of the parameters $\gamma^*$ and the mass over the luminosity ratio. The outcome of the fits is given in Appendix D. One can notice that the fitting results for the ratio $M/L$ turned out to be close to 1 consistently with the absence of dark matter.

Finally, using the observational Tully–Fisher relation we obtained the value for the universal parameter $a_0 = (9.35 \pm 2.22) \times 10^{-11} \,\mathrm{m/s^2}$.

As a final remark, our model is based on a very conservative approach to Einstein's theory of gravity, rather than speculative new radical ideas. Indeed, Einstein's theory: (i) does not introduce other degrees of freedom, contrary to Weyl gravity that propagates a ghost instability, (ii) does not modify the classical Newtonian dynamics such as the MOND theory, and (iii) does not introduce other fields into the standard model of particle physics like in models based on dark matter.

In addition, our purely geometrical model is universal as explained in Section 3, and works perfectly for galaxies' groups and clusters too (work in progress). Furthermore, the data relative to the 175 galaxies have been fit with only one single parameter, which further supports the universality claim stated above.

In other words, as stated in the first paragraph of the introduction, in this paper, we tried to understand gravity instead of modifying it. In particular, we have here figured out what the correct conformal vacuum at the galactic scale should be in Einstein's conformal gravity.

Finally, we would like to make a comparison with our previous work [1] and a similar geometric approach in [38].

In our seminal paper [1], we made several approximations. In the first place, we coupled a massive particle to a spacetime metric solving the EoM of conformal Einsteins' gravity. Unfortunately, this is not just an approximation but has also relevant theoretical implications. Indeed, the point of solving geometrically the galactic rotation curves' issue is based on the conformal invariance, but in the previous paper [1], it was broken explicitly. The right thing to do is to consider particles conformally coupled to gravity so that the full action, including matter, is conformal invariant. Another relevant problem in [1] is related to the gravitational potential $V$. Indeed, the usual relation between $g_{00}$ and $V$ is not very correct for a non-asymptotical Minkowski spacetime. To obtain the correct effective Newtonian potential, in this paper all the evaluated observables are consistent with the general coordinate invariance (proper time, distances, etc. are all invariant). In particular, we evaluated the velocity of a test particle (a star in the galaxy) without making any approximation and consistently with the diffeomorphism invariance. Only in the end, we made some approximations to end up with a simple handy form of the potential, namely $log r$. It is here interesting to remember how things went during our first project on the geometric origin of the rotation curves. Honestly, at that time we also considered particles conformally coupled to gravity, but we immediately realized that the potential for them was the same as in Newtonian gravity, i.e., $-GM/x$. Hence, we gave up and considered an explicit breaking of the conformal symmetry introducing massive particles. What we did not realize at that time is that, asymptotically, the velocity does not go to zero, but to a constant in an infinite amount of proper time, as proven in this paper. Indeed, it is the singularity in the conformal rescaling to makes every consistent.

In comparison to the previous work, in this paper, we also carefully addressed the following issues. (i) The regularity of the Kretschemann at infinity and in $r = 0$. Indeed, the rescaling of the metric proposed in this paper also takes care of the black hole's singularity. (ii) The geodesic completion of the metric has been carefully investigated for conformally coupled massive particles and massless particles.

In a very interesting paper [38], the authors assume an intrinsic fractal structure of spacetime that implies a modification of Einstein's equations and in the end a modification of the gravitational potential. In our paper, the fundamental theory is Einstein's gravity without any modification and extra new fundamental degrees of freedom. Indeed, it has been known since the 1970s that Einstein's gravity is actually Einstein's conformal gravity in its spontaneously broken conformal phase, namely in the Higgs phase of Weyl's invariance. In our paper, we broke the conformal symmetry spontaneously to a non-trivial vacuum, an exact solution of the EoM of Einstein's conformal gravity, that is not only a spacetime-dependent function but also singular. Such singularity is unattainable by any particle, massive or massless, so that the spacetime is on one side geodetically complete, and the other side provides an effective confining asymptotic potential consistent with the observations.

**Author Contributions:** Conceptualization, L.M.; Methodology, T.Z.; Software, Q.L.; Formal analysis, L.M., T.Z. and Q.L.; Investigation, L.M., T.Z.; Data curation, T.Z.; Writing—Original draft, L.M. and T.Z.; Writing—Review & editing, L.M.; Supervision, L.M. All authors have read and agreed to the published version of the manuscript.

**Funding:** This work was funded by the Basic Research Program of the Science, Technology, and Innovation Commission of Shenzhen Municipality (grant no. JCYJ20180302174206969).

**Data Availability Statement:** All data are included in this paper. All details of data analysis are available upon request by contact with the authors.

**Acknowledgments:** The authors thank the referees for useful comments.

**Conflicts of Interest:** The authors declare no conflict of interest.

## Appendix A. Radial Geodesic Equations in the Metric (10)

We here derive the radial geodesic equation for massless and conformally coupled particles in the metric (10).

### *Appendix A.1. Massless Particles*

In this section, we derive the radial geodesic equation for light in the metric (10). Since like (7) also (10) is independent of the $t$- and $\varphi$- coordinates, according to (42) the following quantities are conserved,

$$e = -\xi \cdot u = -\hat{g}_{tt}u^t = Q^2(r)\left(1 - \frac{2MQ^2(r)}{r}\right)\frac{dt}{d\lambda} = Q^2(r)\left(1 - \frac{2MQ^2(r)}{r}\right)\dot{t}, \quad \text{(A1)}$$

$$\ell = \eta \cdot u = \eta^\alpha u^\beta \hat{g}_{\alpha\beta} = \hat{g}_{\phi\beta}u^\phi = \hat{g}_{\phi\phi}u^\phi = r^2 \sin^2\theta\, \dot{\varphi}, \quad \text{(A2)}$$

where we introduce the null vector

$$u^\alpha = \frac{dx^\alpha}{d\lambda} \quad \text{(A3)}$$

that satisfies

$$u \cdot u = \hat{g}_{\alpha\beta}\frac{dx^\alpha}{d\lambda}\frac{dx^\beta}{d\lambda} = 0. \quad \text{(A4)}$$

From (A4) in the equatorial plane (i.e., $\theta = \pi/2$), we obtain the following equation

$$-Q^2(r)\left(1 - \frac{2MQ^2(r)}{r}\right)\dot{t}^2 + \frac{\dot{r}^2}{Q^2(r)\left(1 - \frac{2MQ^2(r)}{r}\right)} + r^2\dot{\varphi}^2 = 0.\qquad(A5)$$

Solving (A1) for $\dot{t}$ and (A2) for $\dot{\varphi}$ and replacing the results in (A5), the radial geodesic Equation ($\ell = 0$) reads:

$$-\frac{e^2}{Q(r)^2\left(1 - \frac{2MQ(r)}{r}\right)} + \frac{\dot{r}^2}{Q^2(r)\left(1 - \frac{2MQ^2(r)}{r}\right)} = 0,\qquad(A6)$$

and we therefore obtain the equation

$$|\dot{r}| = e,\qquad(A7)$$

which coincides with (54).

*Appendix A.2. Conformally Coupled Massive Particles*

We here study the radial geodesic equations for conformally coupled particles in the metric (10), namely for the metric in the radial coordinate $r$. The Lagrangian for a conformally coupled particle reads:

$$L_{cp} = -\sqrt{-f^2\phi^2\hat{g}_{\mu\nu}\dot{x}^\mu\dot{x}^\nu},\qquad(A8)$$

and the translation invariance in the time-like coordinate $t$ implies:

$$\frac{\partial L_{cp}}{\partial \dot{t}} = -\frac{f^2\phi^2\hat{g}_{tt}\dot{t}}{L_{cp}} = \text{const.} = -E,\qquad(A9)$$

thus the equation of $\dot{t}$ reads

$$\dot{t} = \frac{L_{cp}E}{f^2\phi^2\hat{g}_{tt}}.\qquad(A10)$$

In the proper time gauge, from the equation $\hat{g}_{\mu\nu}\dot{x}^\mu\dot{x}^\nu = -1$, we have $L_{cp} = -f\phi$ so the equation of $\dot{t}$ become

$$\dot{t} = -\frac{E}{f\phi\,\hat{g}_{tt}}.\qquad(A11)$$

Therefore, plugging (A11) into the equation $\hat{g}_{\mu\nu}\dot{x}^\mu\dot{x}^\nu = -1$, we obtain the differential equation of $r(\tau)$, namely

$$\frac{E^2}{f^2\kappa_4^{-1}Q(r)^{-2}} - \dot{r}^2 = Q(r)^2\left(1 - \frac{2GM}{r}Q(r)\right).\qquad(A12)$$

Defining $e^2 \equiv E^2/(f^2\kappa_4^{-1})$, the equation of $\dot{r}$ is reduced to

$$\dot{r}^2 = Q(r)^2\left(e^2 - 1 + \frac{2GM}{r}Q(r)\right),\qquad(A13)$$

which coincides with (31).

## Appendix B. The Cosmological Constant Is Not an Issue in Our Model

It is commonly accepted that the value of the cosmological constant is non-zero ($\Lambda \sim 10^{-56}$ cm$^{-2}$). Therefore, we will have to more correctly consider the rescaling of the Schwarzschild–de Sitter spacetime instead of (7) or (10), namely

$$d\hat{s}^{*2} = Q^2(x)\left[-\left(1 - \frac{2GM}{c^2 x} - \frac{\Lambda}{3}x^2\right)dt^2 + \frac{dx^2}{1 - \frac{2GM}{x} - \frac{\Lambda}{3}x^2} + x^2\Omega^{(2)}\right], \quad (A14)$$

$$Q(x) = \frac{1}{1 - \frac{\gamma^*}{2}x}, \quad (A15)$$

or in the radial coordinate $r$,

$$d\hat{s}^{*2} = -Q^2(r)\left(1 - \frac{2GMQ(r)}{r} - \frac{\Lambda}{3}\frac{r^2}{Q^2(r)}\right)c^2 dt^2 + \frac{dr^2}{Q^2(r)\left(1 - \frac{2GMQ(r)}{r} - \frac{\Lambda}{3}\frac{r^2}{Q^2(r)}\right)} + r^2 d\Omega^{(2)}, \quad (A16)$$

$$Q(r) = 1 + \frac{\gamma^*}{2}r.$$

Notice that the metric is till in the form $g_{00}(r) = -1/g_{11}(r)$. If we focus on (A16) and we consider the limit $r \gg 2GM$ together with the approximation $GM\gamma^* \ll 1$, the metric (A16) simplifies to:

$$\begin{aligned}
d\hat{s}^{*2} &\approx -\left(Q^2(r) - \frac{\Lambda}{3}r^2\right)dt^2 + \frac{dr^2}{\left(Q^2(r) - \frac{\Lambda}{3}r^2\right)} + r^2 d\Omega^{(2)} \\
&= -\left(1 + \gamma^* r + \frac{\gamma^{*2}}{4}r^2 - \frac{\Lambda}{3}r^2\right)dt^2 + \frac{dr^2}{\left(1 + \gamma^* r + \frac{\gamma^{*2}}{4}r^2 - \frac{\Lambda}{3}r^2\right)} + r^2 d\Omega^{(2)}.
\end{aligned} \quad (A17)$$

However, since $\gamma^{*2} \gg \Lambda$ (we will see later that $\gamma^* \sim 10^{-21}$ m$^{-1}$) then the presence of the cosmological constant will not affect our analysis[4]. In other words, the cosmological constant present in the action and fixed by the observations at the Hubble's scale does not affect the physics at the galactic scale.

Finally, we want to make the following speculative comment about the potential impact of $\gamma^*$ on the physics at the Hubble's scale. It deserves to be noticed that the value of the radius of de Sitter's spacetime (proportional to the inverse of the square root of the cosmological constant) is about the radius of the Universe. Therefore, for $\ell_c$ comparable to the radius of the Universe, the two contributions quadratic in $r$ in (A17) could in principle cancel each other for a proper choice of $\gamma^*$. However, this is not the case as we are going to show. We remember that $\sqrt{\Lambda} \simeq 1.05 \times 10^{-26}$ m$^{-1}$. Using a formula that we will derive later and the value of $a_0$, which we will obtain from the fit of the data, the value of $\gamma^*$ for the mass of the all Universe is $\gamma^* = 6.11 \times 10^{-27}$. Finally, the monomial proportional to $r^2$ in $g_{tt}$ and $g_{rr}$ reads:

$$\frac{\gamma^{*2}}{4} - \frac{\Lambda}{3} \simeq \frac{3.74 \times 10^{-53}\,\text{m}^{-2}}{4} - \frac{1.10 \times 10^{-52}\,\text{m}^{-2}}{3} \simeq -2.73 \times 10^{-53}. \quad (A18)$$

Therefore, $\gamma^*$ cannot change the sign of the cosmological constant, and the physics at a large scale is still described by the de Sitter metric. To compare the two contributions proportional to $r^2$, we introduce an effective cosmological constant for AdS, namely

$$|\Lambda_{\gamma^*}| = 3 \times \frac{\gamma^{*2}}{4} = 2.81 \times 10^{-53}\,\text{m}^{-2}, \quad (A19)$$

which we have to compare with $\Lambda$ to finally obtain the following ratio,

$$\frac{\Lambda}{|\Lambda_{\gamma^*}|} \simeq 3.91. \quad (A20)$$

Notice that there is no fine-tuning between the values of $\gamma^*$ and $\Lambda$. Indeed, $\gamma^*$ is a result of this paper, which will be derived later by comparing our model with the observations, while for $\Lambda$ we used the observed value.

Therefore, according to the discussion above one might be led to change the value of the cosmological constant in the action to take into account the $\gamma^*$ contribution. However, we have to carefully pay attention to the right matter content in the whole Universe. Indeed, at the cosmological scale, the energy-momentum tensor used in Einstein's EoM is one of a perfect fluid regardless of the interaction between the compact objects spread in the Universe. Therefore, the solution of Einstein's EoM is not affected by $\gamma^*$ as well as is not affected by the Schwarzschild–de Sitter geometry[5] surrounding the point-like masses that fill the whole Universe in Einstein's gravity when the conformal symmetry is not taken into account.

We can summarize the content of this section in two main statements: (i) the observed value of the cosmological constant does not affect our model at the galactic scale because $\gamma^{*2} \gg \Lambda$, and (ii) the value of $\gamma^*$ evaluated at the mass of the whole Universe does not affect the physics at the Hubble's scale because the latter one is well described by the FRW metric for a perfect fluid regardless of the gravitational interaction between masses in the Universe.

**Appendix C. Galactic Parameters For 175 Galaxies**

In this section, we remember the main data from the SPARC database [37] needs the fits of the square velocity (114), and we list the values of $\gamma^*$ and $M/L$ that turn out from our fits for 175 galaxies.

**Table A1.** Galactic parameters of the 175 galaxy samples.

| Galaxy Name | Hubble Type | Distance (Mpc) | $L$ ($10^9 L_\odot$) | $R_0$ (kpc) | $M_{HI}$ ($10^9 M_\odot$) | $(M/L)_{stars}$ ($M_\odot/L_\odot$) | $\gamma^*$ (kpc$^{-1}$) |
|---|---|---|---|---|---|---|---|
| CamB | Im | 3.36 | 0.075 | 0.47 | 0.012 | 0.0883 | 8.94 |
| D512-2 | Im | 15.2 | 0.325 | 1.24 | 0.081 | 1.75 | 0.556 |
| D564-8 | Im | 8.79 | 0.033 | 0.61 | 0.029 | 0.295 | 10.6 |
| D631-7 | Im | 7.72 | 0.196 | 0.7 | 0.29 | 1.19 | 4.13 |
| DDO064 | Im | 6.8 | 0.157 | 0.69 | 0.211 | 0.411 | 6.18 |
| DDO154 | Im | 4.04 | 0.053 | 0.37 | 0.275 | 1.09 | 0.6 |
| DDO161 | Im | 7.5 | 0.548 | 1.22 | 1.378 | 0.0892 | 0.571 |
| DDO168 | Im | 4.25 | 0.191 | 1.02 | 0.413 | 1.01 | 3.32 |
| DDO170 | Im | 15.4 | 0.543 | 1.95 | 0.735 | 1.03 | 1.36 |
| ESO079-G014 | Sbc | 28.7 | 51.733 | 5.08 | 3.14 | 0.657 | 0.311 |
| ESO116-G012 | Sd | 13 | 4.292 | 1.51 | 1.083 | 0.676 | 1.08 |
| ESO444-G084 | Im | 4.83 | 0.071 | 0.46 | 0.135 | 1.24 | 7.35 |
| ESO563-G021 | Sbc | 60.8 | 311.177 | 5.45 | 24.298 | 0.56 | 0.139 |
| F561-1 | Sm | 66.4 | 4.077 | 2.79 | 1.622 | 0.2 | 0.5 |
| F563-1 | Sm | 48.9 | 1.903 | 3.52 | 3.2 | 2.71 | 0.725 |
| F563-V1 | Im | 54 | 1.54 | 3.79 | 0.61 | 1.17 | 0 |
| F563-V2 | Im | 59.7 | 2.986 | 2.43 | 2.169 | 3.07 | 0.409 |
| F565-V2 | Im | 51.8 | 0.559 | 2.17 | 0.699 | 1.06 | 3.28 |
| F567-2 | Sm | 79 | 2.134 | 3.08 | 2.449 | 0.5 | 0.68 |

**Table A1.** *Cont.*

| Galaxy Name | Hubble Type | Distance (Mpc) | $L$ ($10^9 L_\odot$) | $R_0$ (kpc) | $M_{\mathrm{HI}}$ ($10^9 M_\odot$) | $(M/L)_{\mathbf{stars}}$ ($M_\odot/L_\odot$) | $\gamma^*$ (kpc$^{-1}$) |
|---|---|---|---|---|---|---|---|
| F568-1 | Sc | 90.7 | 6.252 | 5.18 | 4.498 | 4.49 | 0.283 |
| F568-3 | Sd | 82.4 | 8.346 | 4.99 | 3.195 | 1.81 | 0.257 |
| F568-V1 | Sd | 80.6 | 3.825 | 2.85 | 2.491 | 2.64 | 0.362 |
| F571-8 | Sc | 53.3 | 10.164 | 3.56 | 1.782 | 1.099 | 0.662 |
| F571-V1 | Sd | 80.1 | 1.849 | 2.47 | 1.217 | 0.633 | 1.57 |
| F574-1 | Sd | 96.8 | 6.537 | 4.46 | 3.524 | 1.9 | 0.277 |
| F574-2 | Sm | 89.1 | 2.877 | 3.76 | 1.701 | 0.0654 | 0.871 |
| F579-V1 | Sc | 89.5 | 11.848 | 3.37 | 2.245 | 1.4 | 0.205 |
| F583-1 | Sm | 35.4 | 0.986 | 2.36 | 2.126 | 1.51 | 0.922 |
| F583-4 | Sc | 53.3 | 1.715 | 1.93 | 0.641 | 0.955 | 0.736 |
| IC2574 | Sm | 3.91 | 1.016 | 2.78 | 1.036 | 0.319 | 2.43 |
| IC4202 | Sbc | 100.4 | 179.749 | 4.78 | 12.326 | 0.64 | 0.107 |
| KK98-251 | Im | 6.8 | 0.085 | 1.34 | 0.115 | 0.789 | 8.4 |
| NGC0024 | Sc | 7.3 | 3.889 | 1.34 | 0.676 | 1.49 | 0.542 |
| NGC0055 | Sm | 2.11 | 4.628 | 6.11 | 1.565 | 2.67 | 0.329 |
| NGC0100 | Scd | 13.5 | 3.232 | 1.66 | 1.99 | 0.345 | 1.07 |
| NGC0247 | Sd | 3.7 | 7.332 | 3.74 | 1.746 | 1 | 0.489 |
| NGC0289 | Sbc | 20.8 | 72.065 | 6.74 | 27.469 | 1.122 | 0.0833 |
| NGC0300 | Sd | 2.08 | 2.922 | 1.75 | 0.936 | 0.612 | 1.35 |
| NGC0801 | Sc | 80.7 | 312.57 | 8.72 | 23.201 | 0.847 | 0.04 |
| NGC0891 | Sb | 9.91 | 138.34 | 2.55 | 4.462 | 0.532 | 0.0876 |
| NGC1003 | Scd | 11.4 | 6.82 | 1.61 | 5.88 | 0.121 | 0.271 |
| NGC1090 | Sbc | 37 | 72.045 | 3.53 | 8.783 | 0.513 | 0.147 |
| NGC1705 | BCD | 5.73 | 0.533 | 0.39 | 0.139 | 1.22 | 2.09 |
| NGC2366 | Im | 3.27 | 0.236 | 0.65 | 0.647 | 0.15 | 0.35 |
| NGC2403 | Scd | 3.16 | 10.041 | 1.39 | 3.199 | 0.446 | 0.646 |
| NGC2683 | Sb | 9.81 | 80.415 | 2.18 | 1.406 | 0.559 | 0.165 |
| NGC2841 | Sb | 14.1 | 188.121 | 3.64 | 9.775 | 0.822 | 0.158 |
| NGC2903 | Sbc | 6.6 | 81.863 | 2.33 | 2.552 | 0.684 | 0.126 |
| NGC2915 | BCD | 4.06 | 0.641 | 0.55 | 0.508 | 0.286 | 3.05 |
| NGC2955 | Sb | 97.9 | 319.422 | 18.76 | 28.949 | 3.3 | 0.0396 |
| NGC2976 | Sc | 3.58 | 3.371 | 1.01 | 0.172 | 0.498 | 1.33 |
| NGC2998 | Sc | 68.1 | 150.902 | 6.2 | 23.451 | 1 | 0.0576 |
| NGC3109 | Sm | 1.33 | 0.194 | 1.56 | 0.477 | 0.854 | 5.56 |
| NGC3198 | Sc | 13.8 | 38.279 | 3.14 | 10.869 | 0.533 | 0.179 |
| NGC3521 | Sbc | 7.7 | 84.836 | 2.4 | 4.154 | 0.808 | 0.768 |
| NGC3726 | Sc | 18 | 70.234 | 3.4 | 6.473 | 0.315 | 0.292 |
| NGC3741 | Im | 3.21 | 0.028 | 0.2 | 0.182 | 0.638 | 0.915 |
| NGC3769 | Sb | 18 | 18.679 | 3.38 | 5.529 | 1.2 | 0.0986 |
| NGC3877 | Sc | 18 | 72.535 | 2.53 | 1.483 | 0.33 | 0.288 |
| NGC3893 | Sc | 18 | 58.525 | 2.38 | 5.799 | 0.668 | 0.143 |
| NGC3917 | Scd | 18 | 21.966 | 2.63 | 1.888 | 0.47 | 0.523 |
| NGC3949 | Sbc | 18 | 38.067 | 3.59 | 3.371 | 1.74 | 0 |
| NGC3953 | Sbc | 18 | 141.301 | 4.89 | 2.832 | 1.08 | 0 |
| NGC3972 | Sbc | 18 | 14.353 | 2.18 | 1.214 | 0.545 | 0.647 |
| NGC3992 | Sbc | 23.7 | 226.932 | 4.96 | 16.599 | 0.574 | 0.117 |
| NGC4010 | Sd | 18 | 17.193 | 2.81 | 2.832 | 0.465 | 0.591 |
| NGC4013 | Sb | 18 | 79.094 | 3.53 | 2.967 | 0.632 | 0.154 |
| NGC4051 | Sbc | 18 | 95.268 | 4.65 | 2.697 | 7.62 | 0 |
| NGC4068 | Im | 4.37 | 0.236 | 0.59 | 0.154 | 0.118 | 5.04 |
| NGC4085 | Sc | 18 | 21.724 | 1.65 | 1.349 | 0.367 | 0.526 |
| NGC4088 | Sbc | 18 | 107.286 | 2.58 | 8.226 | 0.227 | 0.236 |
| NGC4100 | Sbc | 18 | 59.394 | 2.15 | 3.102 | 0.431 | 0.318 |
| NGC4138 | S0 | 18 | 44.111 | 1.51 | 1.483 | 0.574 | 0.223 |
| NGC4157 | Sb | 18 | 105.62 | 2.32 | 8.226 | 0.277 | 0.238 |
| NGC4183 | Scd | 18 | 10.838 | 2.79 | 3.506 | 0.981 | 0.231 |
| NGC4214 | Im | 2.87 | 1.141 | 0.51 | 0.486 | 0.844 | 1.02 |

**Table A1.** *Cont.*

| Galaxy Name | Hubble Type | Distance (Mpc) | $L$ ($10^9 L_\odot$) | $R_0$ (kpc) | $M_{HI}$ ($10^9 M_\odot$) | $(M/L)_{stars}$ ($M_\odot/L_\odot$) | $\gamma^*$ ($kpc^{-1}$) |
|---|---|---|---|---|---|---|---|
| NGC4217 | Sb | 18 | 85.299 | 2.94 | 2.562 | 0.461 | 0.205 |
| NGC4389 | Sbc | 18 | 21.328 | 2.79 | 0.539 | 0.367 | 0.53 |
| NGC4559 | Scd | 9 | 19.377 | 2.1 | 5.811 | 0.356 | 0.259 |
| NGC5005 | Sbc | 16.9 | 178.72 | 9.45 | 1.28 | 4.63 | 0 |
| NGC5033 | Sc | 15.7 | 110.509 | 5.16 | 11.314 | 1.03 | 0.0951 |
| NGC5055 | Sbc | 9.9 | 152.922 | 3.2 | 11.722 | 0.458 | 0.0863 |
| NGC5371 | Sbc | 39.7 | 340.393 | 7.44 | 11.18 | 0.593 | 0.0542 |
| NGC5585 | Sd | 7.06 | 2.943 | 1.53 | 1.683 | 0.51 | 0.939 |
| NGC5907 | Sc | 17.3 | 175.425 | 5.34 | 21.025 | 0.699 | 0.0755 |
| NGC5985 | Sb | 39.7 | 208.728 | 7.01 | 11.586 | 1.32 | 0.0692 |
| NGC6015 | Scd | 17 | 32.129 | 2.3 | 5.834 | 0.609 | 0.261 |
| NGC6195 | Sb | 127.8 | 391.076 | 13.94 | 20.907 | 1.35 | 0.0506 |
| NGC6503 | Scd | 6.26 | 12.845 | 2.16 | 1.744 | 0.931 | 0.279 |
| NGC6674 | Sb | 51.2 | 214.654 | 6.04 | 32.165 | 0.892 | 0.063 |
| NGC6789 | BCD | 3.52 | 0.1 | 0.31 | 0.017 | 1.67 | 7.75 |
| NGC6946 | Scd | 5.52 | 66.173 | 2.44 | 5.67 | 0.533 | 0.123 |
| NGC7331 | Sb | 14.7 | 250.631 | 5.02 | 11.067 | 0.659 | 0.0632 |
| NGC7793 | Sd | 3.61 | 7.05 | 1.21 | 0.861 | 0.594 | 0.609 |
| NGC7814 | Sab | 14.4 | 74.529 | 2.54 | 1.07 | 1.04 | 0.116 |
| PGC51017 | BCD | 13.6 | 0.155 | 0.53 | 0.201 | 0 | 0 |
| UGC00128 | Sdm | 64.5 | 12.02 | 5.95 | 7.431 | 1.35 | 0.288 |
| UGC00191 | Sm | 17.1 | 2.004 | 1.58 | 1.343 | 1.08 | 0.706 |
| UGC00634 | Sm | 30.9 | 2.989 | 2.45 | 3.663 | 0.748 | 0.774 |
| UGC00731 | Im | 12.5 | 0.323 | 2.3 | 1.807 | 3.08 | 1.24 |
| UGC00891 | Sm | 10.2 | 0.374 | 1.43 | 0.428 | 0.464 | 4.01 |
| UGC01230 | Sm | 53.7 | 7.62 | 4.34 | 6.43 | 2.965 | 0.0649 |
| UGC01281 | Sdm | 5.27 | 0.353 | 1.63 | 0.294 | 0.915 | 4.12 |
| UGC02023 | Im | 10.4 | 1.308 | 1.55 | 0.477 | 0.346 | 2.14 |
| UGC01281 | Sdm | 5.27 | 0.353 | 1.63 | 0.294 | 0.915 | 4.12 |
| UGC02023 | Im | 10.4 | 1.308 | 1.55 | 0.477 | 0.346 | 2.14 |
| UGC02259 | Sdm | 10.5 | 1.725 | 1.62 | 0.494 | 2.19 | 0.538 |
| UGC02455 | Im | 6.92 | 3.649 | 0.99 | 0.803 | 0.0341 | 1.28 |
| UGC02487 | S0 | 69.1 | 489.955 | 7.89 | 17.963 | 1.12 | 0.0508 |
| UGC02885 | Sc | 80.6 | 403.525 | 11.4 | 40.075 | 1.18 | 0.0352 |
| UGC02916 | Sab | 65.4 | 124.153 | 6.15 | 23.273 | 1.33 | 0.0635 |
| UGC02953 | Sab | 16.5 | 259.518 | 3.55 | 7.678 | 0.55 | 0.15 |
| UGC03205 | Sab | 50 | 113.642 | 3.19 | 9.677 | 0.659 | 0.154 |
| UGC03546 | Sa | 28.7 | 101.336 | 3.79 | 2.675 | 0.6 | 0.152 |
| UGC03580 | Sa | 20.7 | 13.266 | 2.43 | 4.37 | 0.886 | 0.194 |
| UGC04278 | Sd | 9.51 | 1.307 | 2.21 | 1.116 | 0.882 | 2.14 |
| UGC04305 | Im | 3.45 | 0.736 | 1.16 | 0.69 | 0.134 | 0.824 |
| UGC04325 | Sm | 9.6 | 2.026 | 1.86 | 0.678 | 2.64 | 0.497 |
| UGC04483 | Im | 3.34 | 0.013 | 0.18 | 0.032 | 0.0444 | 6.2 |
| UGC04499 | Sdm | 12.5 | 1.552 | 1.73 | 1.1 | 0.847 | 0.971 |
| UGC05005 | Im | 53.7 | 4.1 | 3.2 | 3.093 | 0.359 | 0.908 |
| UGC05253 | Sab | 22.9 | 171.582 | 8.07 | 16.396 | 1.3 | 0.04 |
| UGC05414 | Im | 9.4 | 1.123 | 1.47 | 0.574 | 0.479 | 2.31 |
| UGC05716 | Sm | 21.3 | 0.588 | 1.14 | 1.094 | 0.923 | 0.922 |
| UGC05721 | Sd | 6.18 | 0.531 | 0.38 | 0.562 | 0.722 | 1.4 |
| UGC05750 | Sdm | 58.7 | 3.336 | 3.46 | 1.099 | 0.409 | 1.34 |
| UGC05764 | Im | 7.47 | 0.085 | 1.17 | 0.163 | 6.58 | 2.42 |
| UGC05829 | Im | 8.64 | 0.564 | 1.99 | 1.023 | 0.624 | 2.63 |
| UGC05918 | Im | 7.66 | 0.233 | 1.66 | 0.297 | 2.3 | 1.46 |
| UGC05986 | Sm | 8.63 | 4.695 | 1.67 | 2.667 | 0.824 | 0.725 |
| UGC05999 | Im | 47.7 | 3.384 | 3.22 | 2.022 | 0.577 | 1.33 |
| UGC06399 | Sm | 18 | 2.296 | 2.05 | 0.674 | 0.748 | 1.62 |

**Table A1.** *Cont.*

| Galaxy Name | Hubble Type | Distance (Mpc) | $L$ $(10^9 L_\odot)$ | $R_0$ (kpc) | $M_{HI}$ $(10^9 M_\odot)$ | $(M/L)_{stars}$ $(M_\odot/L_\odot)$ | $\gamma^*$ $(\mathrm{kpc}^{-1})$ |
|---|---|---|---|---|---|---|---|
| UGC06446 | Sd | 12 | 0.988 | 1.49 | 1.379 | 1.91 | 0.711 |
| UGC06614 | Sa | 88.7 | 124.35 | 5.1 | 21.888 | 0.434 | 0.142 |
| UGC06628 | Sm | 15.1 | 3.739 | 2.82 | 1.5 | 0.37 | 0.366 |
| UGC06667 | Scd | 18 | 1.397 | 5.15 | 0.809 | 7.52 | 0.598 |
| UGC06786 | S0 | 29.3 | 73.407 | 3.6 | 5.03 | 1.32 | 0.0985 |
| UGC06787 | Sab | 21.3 | 98.256 | 5.37 | 5.03 | 2.01 | 0.0713 |
| UGC06818 | Sm | 18 | 1.588 | 1.39 | 1.079 | 0.157 | 1.94 |
| UGC06917 | Sm | 18 | 6.832 | 2.76 | 2.023 | 1.18 | 0.39 |
| UGC06923 | Im | 18 | 2.89 | 1.44 | 0.809 | 0.48 | 1.44 |
| UGC06930 | Sd | 18 | 8.932 | 3.94 | 3.237 | 1.49 | 0.237 |
| UGC06973 | Sab | 18 | 53.87 | 1.07 | 1.753 | 0.295 | 0.334 |
| UGC06983 | Scd | 18 | 5.298 | 3.21 | 2.967 | 1.97 | 0.307 |
| UGC07089 | Sdm | 18 | 3.585 | 2.26 | 1.214 | 0.333 | 1.29 |
| UGC07125 | Sm | 19.8 | 2.712 | 3.38 | 4.629 | 1.42 | 0.0337 |
| UGC07151 | Scd | 6.87 | 2.284 | 1.25 | 0.616 | 0.691 | 0.843 |
| UGC07232 | Im | 2.83 | 0.113 | 0.29 | 0.046 | 0.57 | 7.29 |
| UGC07261 | Sdm | 13.1 | 1.753 | 1.2 | 1.388 | 0.827 | 0.534 |
| UGC07323 | Sdm | 8 | 4.109 | 2.26 | 0.722 | 0.566 | 1.21 |
| UGC07399 | Sdm | 8.43 | 1.156 | 1.64 | 0.745 | 4.09 | 0.658 |
| UGC07524 | Sm | 4.74 | 2.436 | 3.46 | 1.779 | 1.96 | 0.453 |
| UGC07559 | Im | 4.97 | 0.109 | 0.58 | 0.169 | 0.166 | 2.95 |
| UGC07577 | Im | 2.59 | 0.045 | 0.9 | 0.044 | 0.172 | 11.5 |
| UGC07603 | Sd | 4.7 | 0.376 | 0.53 | 0.258 | 0.438 | 3.75 |
| UGC07608 | Im | 8.21 | 0.264 | 1.5 | 0.535 | 1.16 | 4.77 |
| UGC07690 | Im | 8.11 | 0.858 | 0.57 | 0.39 | 0.692 | 0.763 |
| UGC07866 | Im | 4.57 | 0.124 | 0.61 | 0.118 | 0.56 | 2.6 |
| UGC08286 | Scd | 6.5 | 1.255 | 1.05 | 0.642 | 0.876 | 1.48 |
| UGC07866 | Im | 4.57 | 0.124 | 0.61 | 0.118 | 0.56 | 2.6 |
| UGC08286 | Scd | 6.5 | 1.255 | 1.05 | 0.642 | 0.876 | 1.48 |
| UGC08490 | Sm | 4.65 | 1.017 | 0.67 | 0.72 | 0.999 | 0.854 |
| UGC08550 | Sd | 6.7 | 0.289 | 0.45 | 0.288 | 0.47 | 1.74 |
| UGC08699 | Sab | 39.3 | 50.302 | 3.09 | 3.738 | 1.23 | 0.0997 |
| UGC08837 | Im | 7.21 | 0.501 | 1.72 | 0.32 | 0.462 | 3.42 |
| UGC09037 | Scd | 83.6 | 68.614 | 4.28 | 19.078 | 0.335 | 0.137 |
| UGC09133 | Sab | 57.1 | 282.926 | 6.97 | 33.428 | 0.75 | 0.058 |
| UGC09992 | Im | 10.7 | 0.336 | 1.04 | 0.318 | 0.643 | 1.22 |
| UGC10310 | Sm | 15.2 | 1.741 | 1.8 | 1.196 | 1.24 | 0.536 |
| UGC11455 | Scd | 78.6 | 374.322 | 5.93 | 13.335 | 0.415 | 0.127 |
| UGC11557 | Sdm | 24.2 | 12.101 | 2.75 | 2.605 | 0.215 | 0.704 |
| UGC11820 | Sm | 18.1 | 0.97 | 2.08 | 1.977 | 1.58 | 0.718 |
| UGC11914 | Sab | 16.9 | 150.028 | 2.44 | 0.888 | 0.907 | 0.577 |
| UGC12506 | Scd | 100.6 | 139.571 | 7.38 | 35.556 | 1.32 | 0.0599 |
| UGC12632 | Sm | 9.77 | 1.301 | 2.42 | 1.744 | 1.58 | 0.62 |
| UGC12732 | Sm | 13.2 | 1.667 | 1.98 | 3.66 | 0.704 | 0.548 |
| UGCA281 | BCD | 5.68 | 0.194 | 1.72 | 0.062 | 13.1 | 0 |
| UGCA442 | Sm | 4.35 | 0.14 | 1.18 | 0.263 | 1.86 | 3.37 |
| UGCA444 | Im | 0.98 | 0.012 | 0.83 | 0.067 | 8.9 | 4.57 |

## Appendix D. Fitting the Galactic Rotation Curves of 175 Galaxies

We hereby provide the fits for the galactic orbital velocity (in km/s) as a function of the physical radial distance (in kpc) for 175 galaxies. In each plot, the blue dots with error bars are the data of the observed galactic rotation velocity, and the red curve is the fitting result. The dashed (blue-)curve represents the Newtonian contribution to the velocity square, namely the first term in (114), while the (yellow-)dot-dashed curve shows only the modification due to the conformal rescaling, namely only the second contribution in (114).

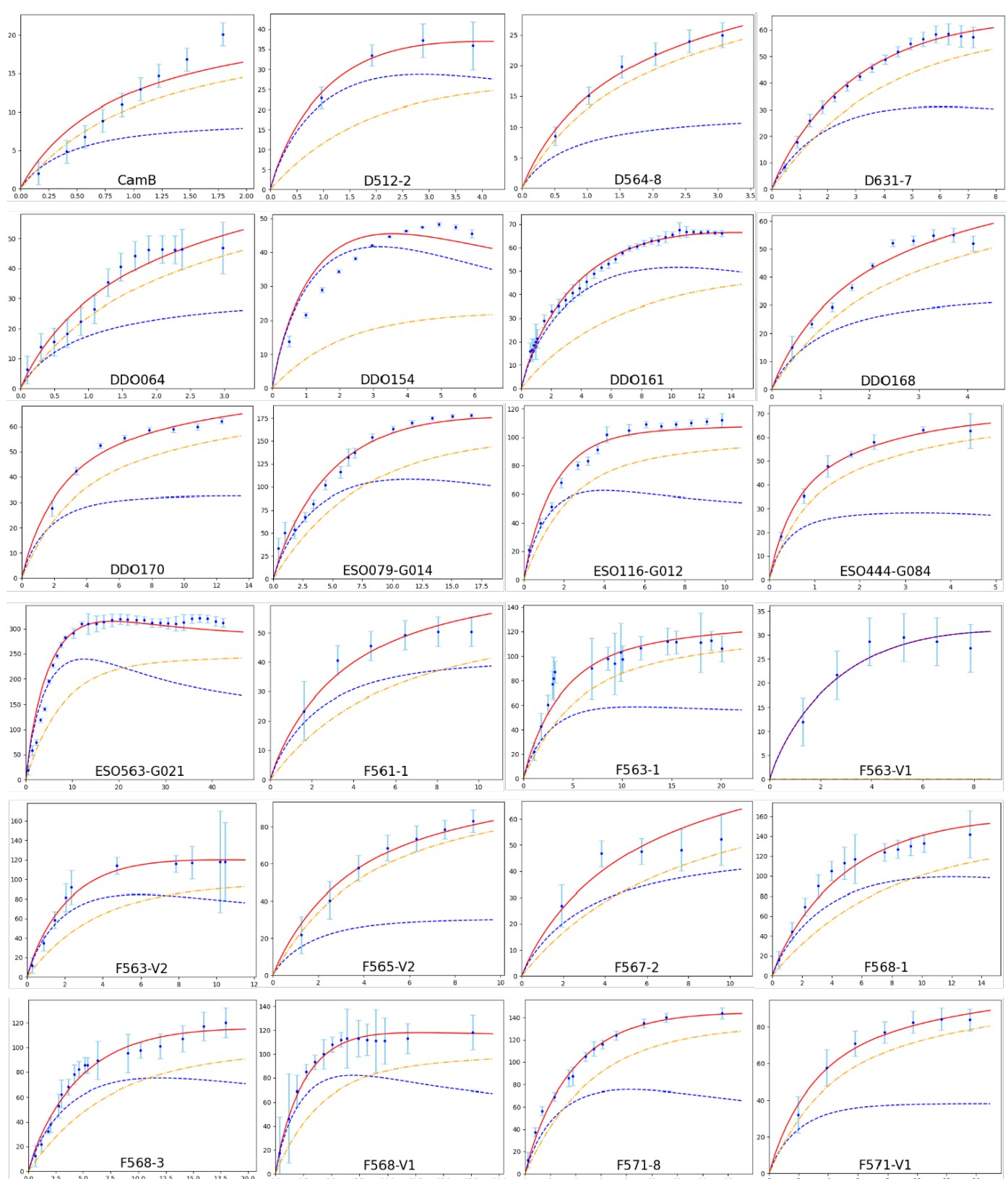

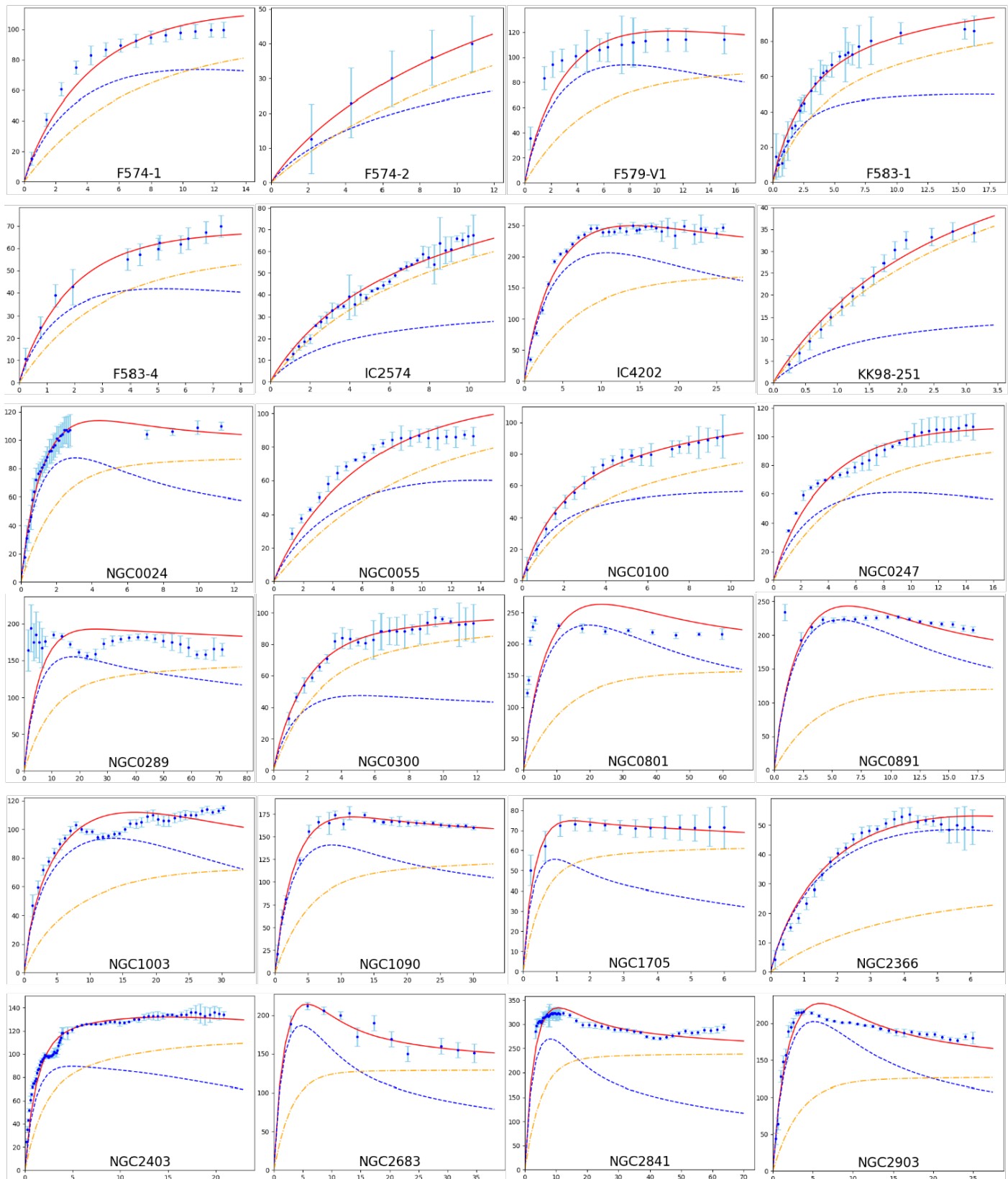

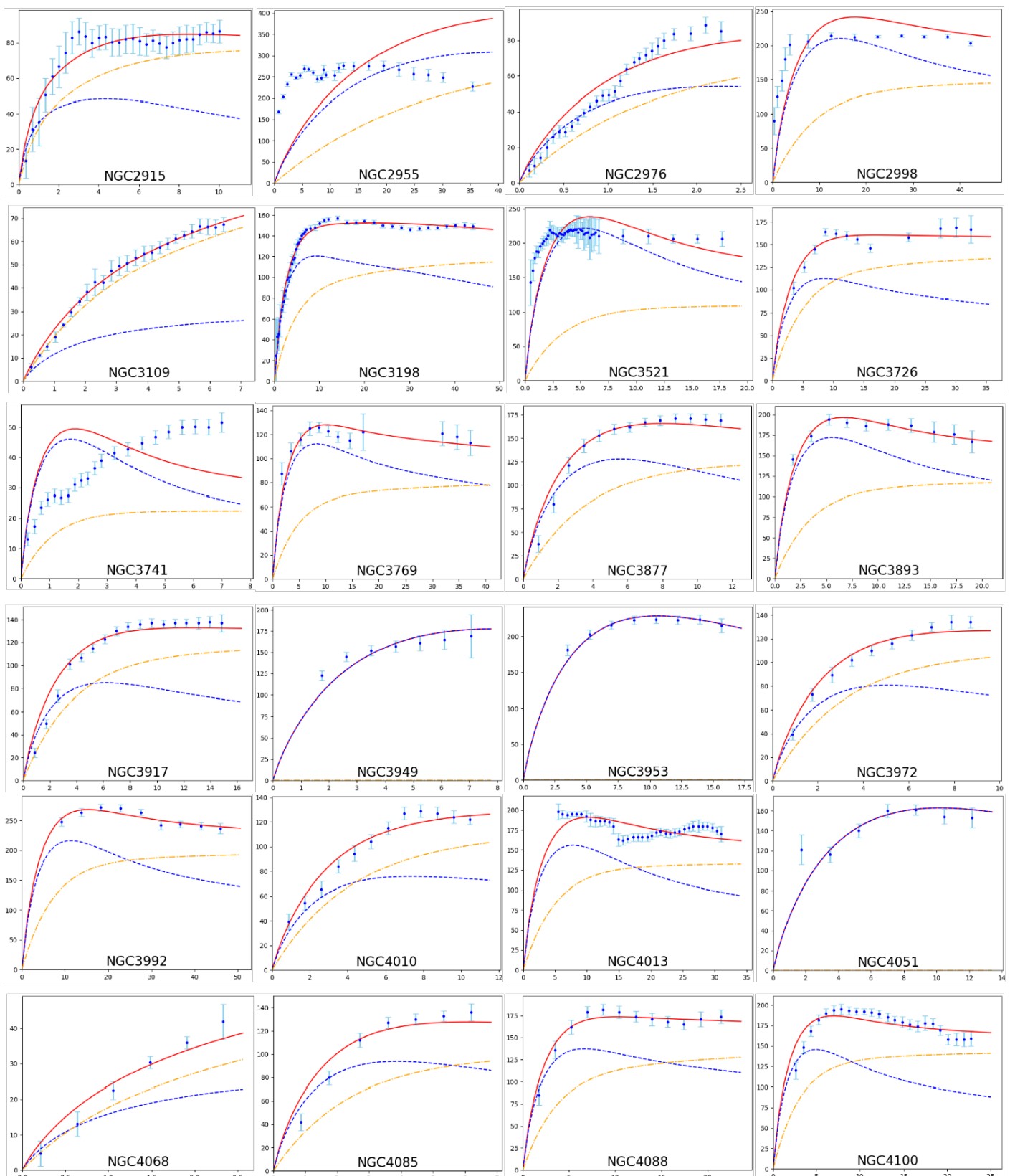

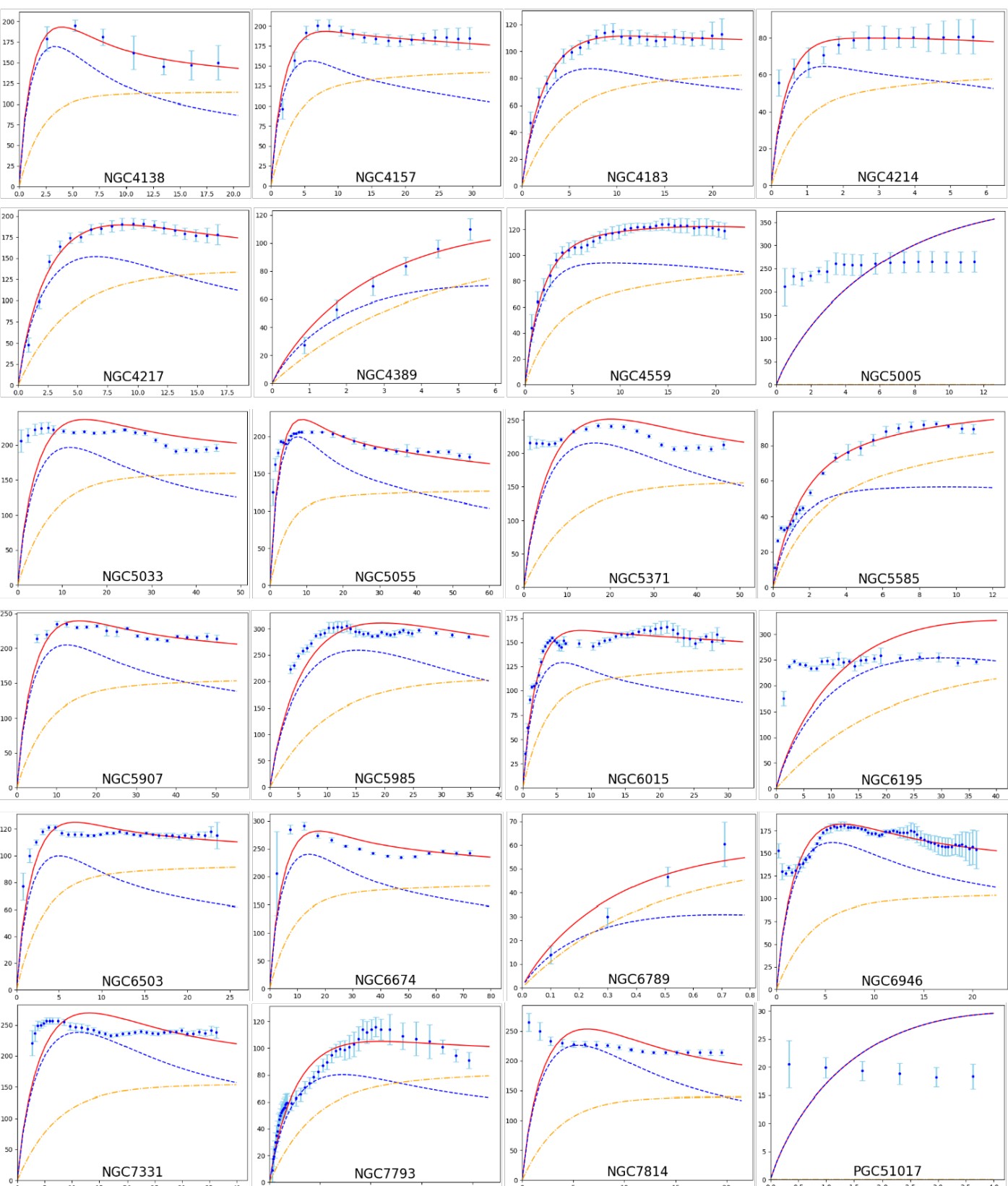

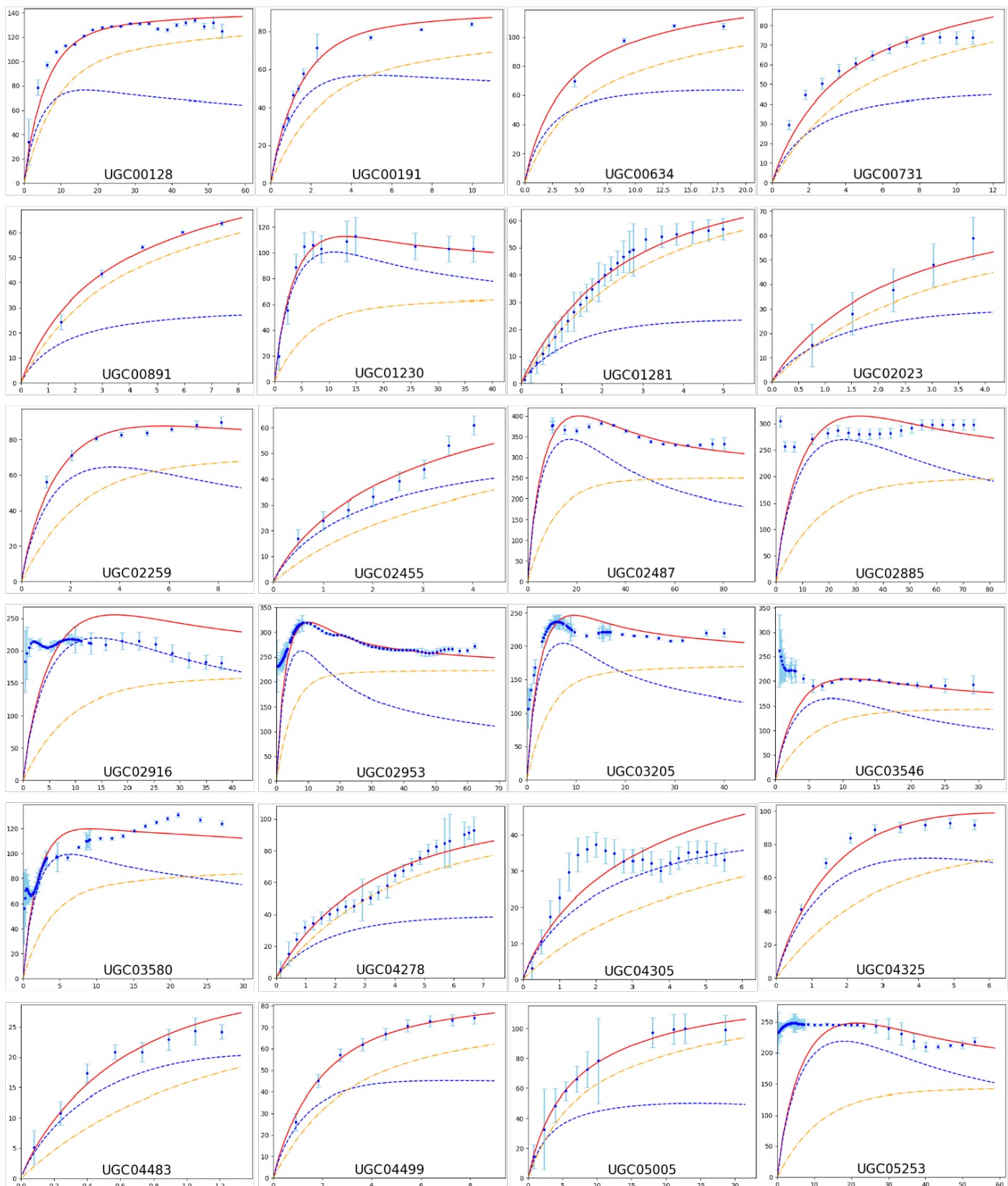

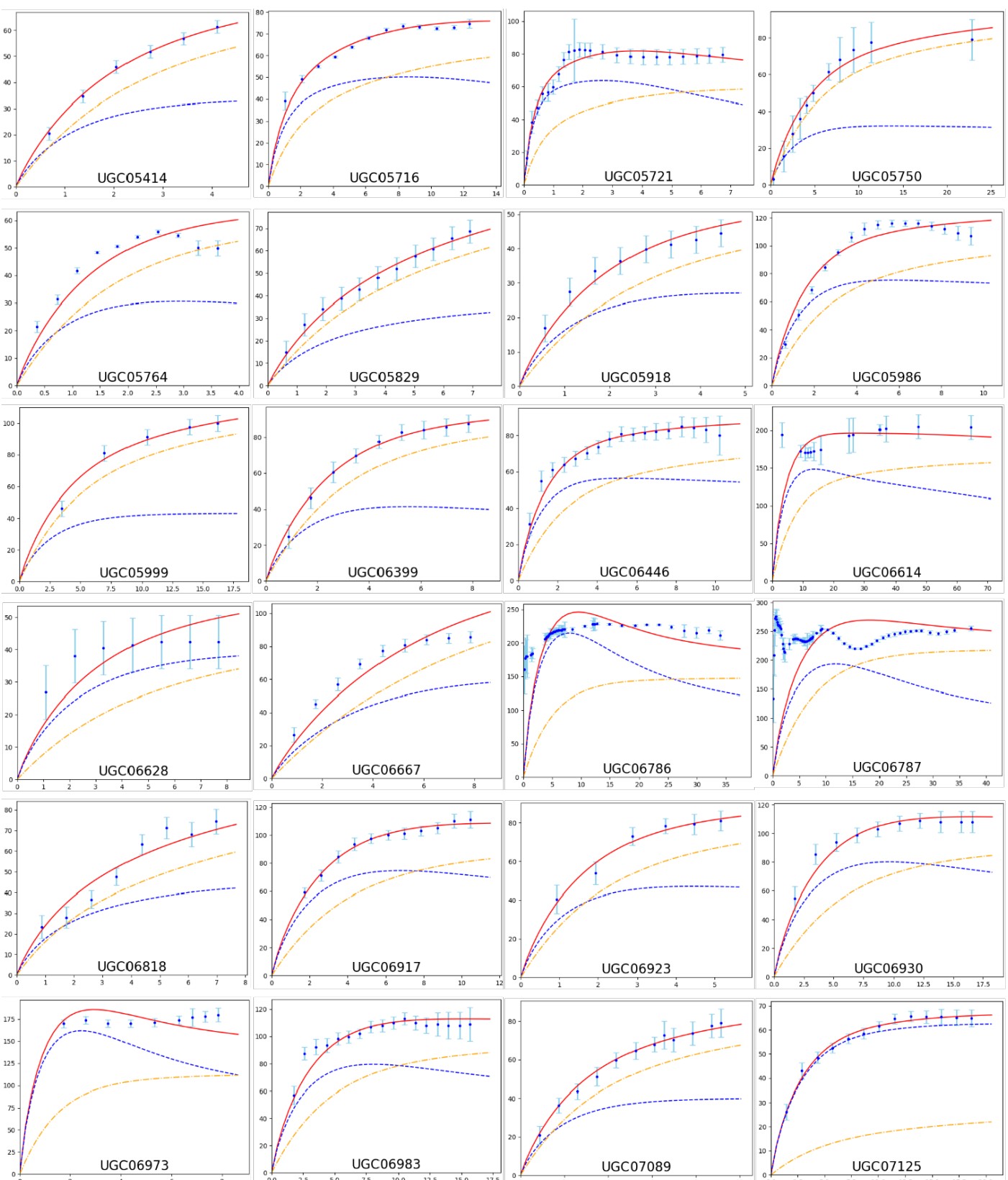

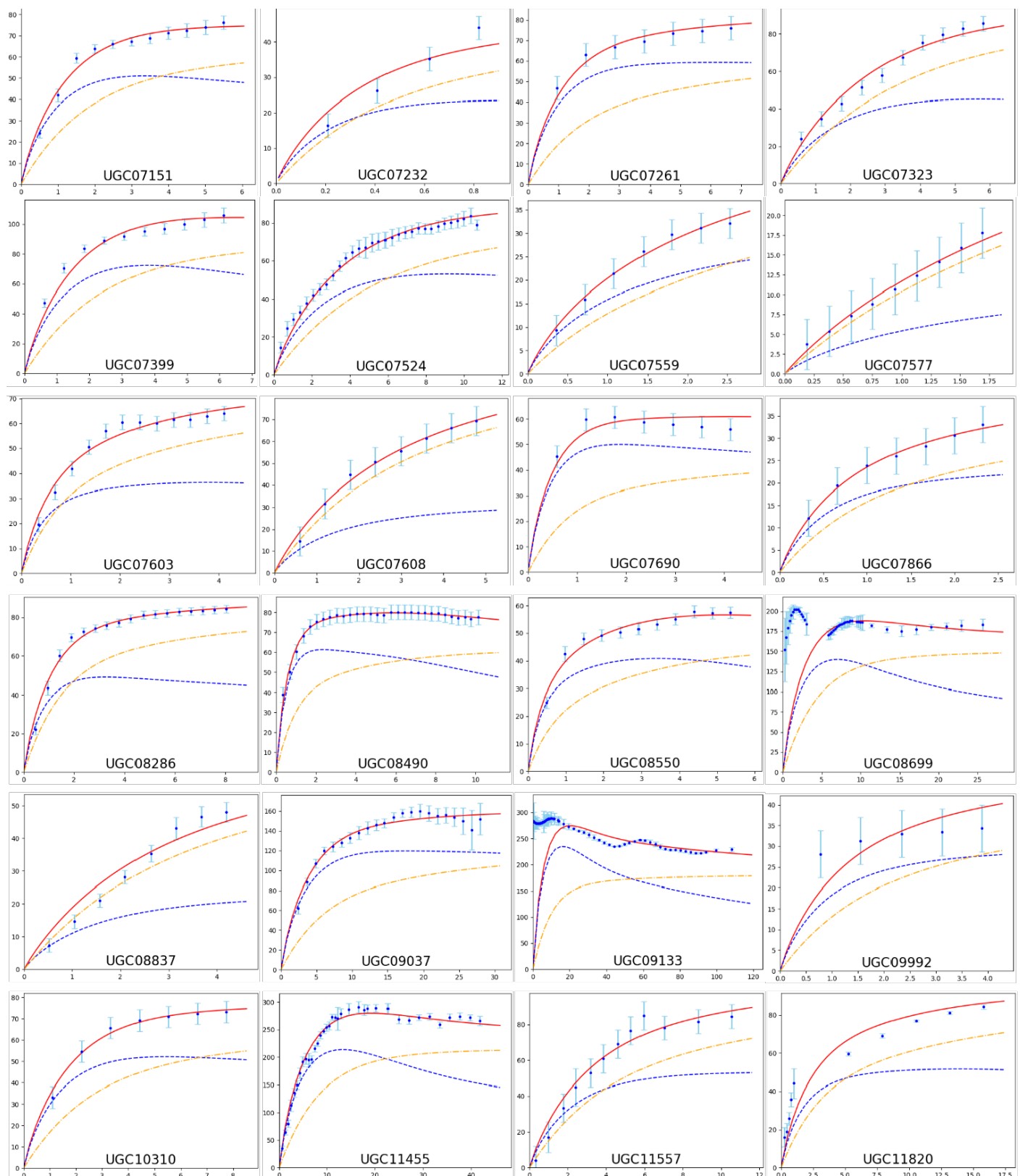

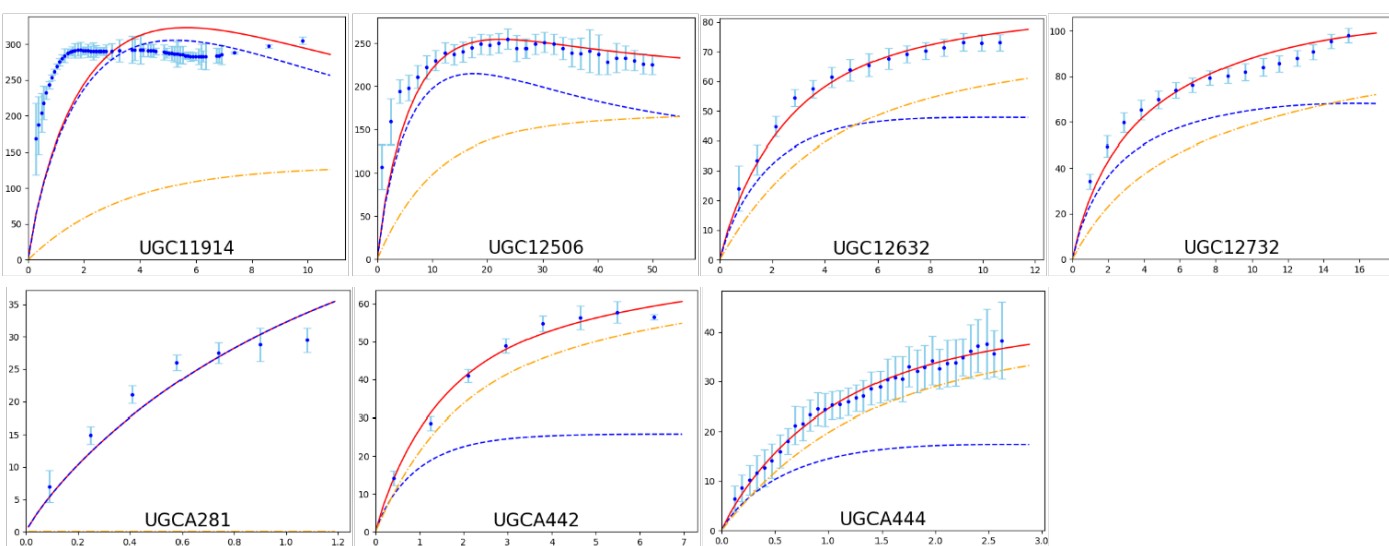

## Notes

[1]    The previous seminal paper [1] did not address the issue of conformally coupled matter that completely changes the geometrical interpretation of our proposal underlining the crucial role of the asymptotic but harmless spacetime singularity. Notice that in [1] the massive particles break explicitly the conformal invariance, even if slightly, making the solution no longer exact. Moreover, we will show in this paper that in the presence of conformally coupled matter we do not need to resort to the global structure of space–time and to invoke the small inhomogeneities on the cosmological scale or the presence of the cosmological constant, which will turn out to be too small to affect on the rotation curves on a galactic scale: "everything will be limited to the single galaxies".

[2]    Notice that a conformal rescaling of the metric does not affect the light bending. Therefore, our model does not suffer from the issue mentioned in [26] for the case of Weyl conformal gravity. Indeed, such a problem is present for exact solutions of Weyl gravity that are not a rescaling of the Schwarzschild metric. We remember that in Weyl gravity we have more solutions because it is a higher derivative theory.

[3]    Notice that assuming the conformal symmetry to be spontaneously broken to $\phi = \kappa_4^{-1}$ and taking the unitary gauge, the action (19) turns into the usual one for a particle with mass $m = f\kappa_4^{-1}$ ($f > 0$). Different values for $f$ provide different mass scales.

[4]    Looking at the Mannheim's paper [32], in the Appendix A5, the potential is defined as usually like $-(g_{00} + 1)/2$ (see the paragraph before formula (A43) and formula (A45) ). However, this is inconsistent with the physical velocity that we obtain from the metric (62). Indeed, the usual derivation of the potential, which one can find for example in Landau's book "Classical Field Theory", does not work for spacetimes not asymptotically flat, which is the case of (62). The correctness of Mannheim's paper lies in the fact that the scales in their model are much larger than the galactic extension. It deserves to be noted that for special values of $\gamma_0$ and $k$ in Mannheim's paper, namely $\gamma_0 = \gamma^*$ and $k^2 = -\gamma^{*2}/4$, the exact solution (62) (it is (5) in Mannheim's paper) of Weyl conformal gravity turns out to be a conformal rescaling of the Minkowski spacetime. A generalization of the model in [32] has been proposed and extensively studied in [39]. The latter paper deals with solutions in $f(T) = T + \alpha T^n$ gravitational theories, where $T$ is the torsion. However, the generality of the results we think will be very useful for improving our model too.

[5]    Notice that the geometry of any compact object in the Universe is not Schwarzschild, but Schwarzschild–de Sitter because of the presence of the cosmological constant in the action as required by the ΛCDM model.

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
