# Peer review of "Geometric Origin of the Galaxies’ Dark Side"

_universe, doi:10.3390/universe10010019_

Round 1
Reviewer 1 Report
Comments and Suggestions for Authors
The authors consider the conformal transformation of the four-dimensional Schwarzschild metric and obtain an exact solution with the scalar field (10) in the conformal Einstein gravity theory. The authors study the regularity of the solution with respect to its curvature invariants and the motions of the massive and massless particles. Then the authors apply their solution without the components of dark matter and dark energy to some galaxy rotation curves and obtain an acceleration parameter (132) that is consistent with some previously obtained values. I recommend the authors to consider the following points:
i) In the reference [1], two of the present authors would have already discussed some galaxy rotation curves in the same background geometry (10). Moreover, such galaxy rotation curves would have been extensively studied in the paper arXiv:2112.13103 [gr-qc]. Then the present authors should discuss in detail the differences between this work and the above papers, including the comparison of their acceleration parameter (132), and emphasize the novelty of the present work.
ii) The term $r^2 \sin ^2 \theta \dot \varphi ^2$ in the equation (A137) would be replaced with the term $r^2 \dot \varphi ^2$.
iii) The authors may include the Penrose diagram of the geometry (10) to easily understand its global structure.
iv) To simplify their manuscripts, the authors may reduce their calculations in the equations (21), (22), (24), (32), (35), (37), (51), (54), (57), (70), (71), (77)-(80), (93), (94), (102), (132), (A138) and (A140)-(A142). Moreover, to clarify their discussions, the authors may move the section 4 on the absence of the cosmological constant in their model to the appendix.
Comments on the Quality of English LanguageThere are a number of typographical errors throughout the present manuscript. For example, the words “vale” in the line 498 and “onsider” in the line 504 would be replaced with the words “value” and “consider”, respectively. Then the authors may carefully review their manuscript and correct the typos.
Author Response
Referee1: Comments and Suggestions for Authors
The authors consider the conformal transformation of the four-dimensional Schwarzschild metric and obtain an exact solution with the scalar field (10) in the conformal Einstein gravity theory. The authors study the regularity of the solution with respect to its curvature invariants and the motions of the massive and massless particles. Then the authors apply their solution without the components of dark matter and dark energy to some galaxy rotation curves and obtain an acceleration parameter (132) that is consistent with some previously obtained values. I recommend the authors to consider the following points:
- i) In the reference [1], two of the present authors would have already discussed some galaxy rotation curves in the same background geometry (10). Moreover, such galaxy rotation curves would have been extensively studied in the paper arXiv:2112.13103 [gr-qc]. Then the present authors should discuss in detail the differences between this work and the above papers, including the comparison of their acceleration parameter (132), and emphasize the novelty of the present work.
Authors: We added the following new paragraph in the conclusions:
Finally, we would like to make a comparison with our previous work \cite{Li:2019ksm} and a similar geometric approach in \cite{Calcagni:2021mmj}.
In our seminal paper \cite{Li:2019ksm}, we made several approximations. In first place we coupled a massive particle to a spacetime metric solving the EoM of conformal Einsteins’ gravity. Unfortunately, this is not just an approximation, but has also relevant theoretical implications. Indeed, the all point of solving geometrically the galactic rotation curves’ issue is based on the conformal invariance, but in the previous paper \cite{Li:2019ksm} it was broken explicitly. The right thing to do is to consider particles conformally coupled to gravity so that the full action, including matter, is conformal invariant. Another relevant problem in \cite{Li:2019ksm} is related to the gravitational potential $V$. Indeed, the usual relation between $g_{00}$ and $V$ is not very correct for a not asymptotically Minkowski spacetime. In order to get the correct effective Newtonian's potential, in this paper all the evaluated observables are consistent with the general coordinate invariance (proper time, distances, etc are all invariant). In particular, we evaluated the velocity of a test particle (a star in the galaxy) without making any approximation and consistently with the Diffeomorphism invariance. Only in the end, we made some approximation to end up with a simple handy form of the potential, namely $log r$.
It is here interesting to remember how things went during our first project on the geometric origin of the rotation curves. Honestly, at that tome we also considered particles conformally coupled to gravity, but we immediately realized that the potential for them was exactly the same as in Newtonian gravity, i.e. $- G M/x$. Hence, we gave up and considered an explicit breaking of the conformal symmetry intruducing massive particles. What we did not realize at that time is that asymptotically the velocity does not go to zero, but to a constant in an infinite ammount of proper time, as proven in this paper. Indeed, it is the singularity in the conformal rescaling to make every consistent.
In comparison to the previous work, in this paper we also carefully addressed the following issues.
(i) The regularity of the Kretschemann at infinity and in $r=0$. Indeed, the rescaling of the metric proposed in this paper also take care of the black hole’s singularity.
(ii) The geodesic completion of the metric has been carefully investigated for conformally coupled massive particles and massless particles.
In the very interesting paper \cite{Calcagni:2021mmj}, the authors assume an intrinsic fractal structure of the spacetime that implies a modification of the Einstein’s equations and in the end a modification of the gravitational potential. In our paper the fundamental theory is Einstein’s gravity without any modification and extra new fundamental degrees of freedom. Indeed, it is known from the $70$s that Einstein’s gravity is actually Einstein’s conformal gravity in its spontaneously broken conformal phase, namely in the Higgs phase of Weyl’s invariance. In our paper we broke the conformal symmetry spontaneously to a non trivial vacuum, exact solution of the EoM of Einstein’s conformal gravity, that is not only a spacetime depended function, but also singular. Such singularity is unattainable by any particle, massive or massless, so that the spacetime is on one side geodetically complete, and on the other side provides and effective confining asymptotic potential consistent with the observations.
Referee1: ii) The term $r^2 \sin ^2 \theta \dot \varphi ^2$ in the equation (A137) would be replaced with the term $r^2 \dot \varphi ^2$.
Authors: We deleted the $sin^2\theta$ in the equation (A137). (Notice that in the new manuscript, the equation (A137) is relabeled as (A132)).
Referee1: iii) The authors may include the Penrose diagram of the geometry (10) to easily understand its global structure.
Authors: We added the following comment in red color at pag.6:
Finally, we notice that the geometry (\ref{Qmetricr}) has the same Penrose diagram of the Schwarzschield black hole because a conformal rescaling can not change the causality structure of the spacetime.
Referee1: iv) To simplify their manuscripts, the authors may reduce their calculations in the equations (21), (22), (24), (32), (35), (37), (51), (54), (57), (70), (71), (77)-(80), (93), (94), (102), (132), (A138) and (A140)-(A142). Moreover, to clarify their discussions, the authors may move the section 4 on the absence of the cosmological constant in their model to the appendix.
Authors: We made some simplifications for the above equations(in red color).
Referee1: Comments on the Quality of English Language: There are a number of typographical errors throughout the present manuscript. For example, the words “vale” in the line 498 and “onsider” in the line 504 would be replaced with the words “value” and “consider”, respectively. Then the authors may carefully review their manuscript and correct the typos.
Authors: We corrected the typos. We thank the referee.

Reviewer 2 Report
Comments and Suggestions for Authors
The idea of applying conformal gravity to solve the cosmological constant problem and the problem of flat galaxy rotation curves was first proposed by Philip D. Mannheim. Camparing with MOND, one of the advantages of the conformal gravity is that it can naturally explain the relation of Milgrom's universal acceleration a0 to Hubble constant H0, i.e., a0~H0c (where c is the speed of light). In this paper, the authors present detailed derivations for the spherically symmetric solutions to the field equations of galaxies in conformal gravity. They then fit the results to 175 SPARC disk galaxies to obtain the value of a0. The results are interesting and significant.
However, there're some problems that need to be resolved.
(1)In the "Abstract" and the main text, the statement "Einstein’s conformal gravity" appears several times, which can be confusing in the context where we emphasize the distinction between Einstein's gravity and the conformal gravity.
(2)The "Introduction" should be rewritten to clearly address the background and main contents of this paper. In particular, the authors should make it clear what the subsequent sections will cover in order to help the readers understand the logical flow and structure of the entire paper.
(3) Too many grammar errors or poor English expressions. ChatGPT are strongly suggested to this end.
Comments on the Quality of English LanguageThere are too many grammar errors or poor English expressions.
For example, "rather then", "more ... then..", "capable to ...", and so on.
In the "Abstract" we read
"we compare our model with a sample of 175 galaxies and we show that our velocity profile very well interpolates the galactic rotation-curves for a proper choice of the only free parameter in the metric and the the mass to luminosity ratios, which turn out to be close to 1 consistently with the absence of dark matter."
It is really difficult to understand。
The first sentence of "Introduction" is
"Despite the enormous successes of the Einstein’s theory of gravity, the latter appears to be about `twenty five percent wrong'. "
If "the latter" refers to "the Einstein's theory of gravity", then what's the "former"?
and so on.
Author Response
Referee2: Comments and Suggestions for Authors
The idea of applying conformal gravity to solve the cosmological constant problem and the problem of flat galaxy rotation curves was first proposed by Philip D. Mannheim. Compared with MOND, one of the advantages of the conformal gravity is that it can naturally explain the relation of Milgrom's universal acceleration a0 to Hubble constant H0, i.e., a0~H0c (where c is the speed of light). In this paper, the authors present detailed derivations for the spherically symmetric solutions to the field equations of galaxies in conformal gravity. They then fit the results to 175 SPARC disk galaxies to obtain the value of a0. The results are interesting and significant.
However, there're some problems that need to be resolved.
(1)In the "Abstract" and the main text, the statement "Einstein’s conformal gravity" appears several times, which can be confusing in the context where we emphasize the distinction between Einstein's gravity and the conformal gravity.
Authors: By Einstein’s conformal gravity we here mean the action (1). The reason to use this name is to distinguish such theory from other conformal invariant theories. Indeed, by conformal gravity usually people mean the Weyl theory $C^2$. However, our results apply to any conformal invariance theory having the Schwarzschild solution as an exact solution.
Referee2: (2)The "Introduction" should be rewritten to clearly address the background and main contents of this paper. In particular, the authors should make it clear what the subsequent sections will cover in order to help the readers understand the logical flow and structure of the entire paper.
Authors: We modified it and added a new paragraph at the end of the Introduction(see red paragraphs on page 3).
Referee2: (3) Too many grammar errors or poor English expressions. ChatGPT are strongly suggested to this end.
Comments on the Quality of English Language
There are too many grammar errors or poor English expressions. For example, "rather then", "more ... then..", "capable to ...", and so on.
Authors: We have corrected these errors and checked the grammar in this paper by Grammarly. We thank the referee for pointing out these errors.
Referee2: In the "Abstract" we read
"we compare our model with a sample of 175 galaxies and we show that our velocity profile very well interpolates the galactic rotation-curves for a proper choice of the only free parameter in the metric and the the mass to luminosity ratios, which turn out to be close to 1 consistently with the absence of dark matter."
It is really difficult to understand。
Authors: We rephrase the sentence:
we compare our model with a sample of 175 galaxies and show that our velocity profile very well interpolates the galactic rotation curves after a proper choice of the only free parameter in the metric. The mass-to-luminosity ratios of galaxies turn out to be close to 1 consistent with the absence of dark matter.
Referee2: The first sentence of "Introduction" is
"Despite the enormous successes of the Einstein’s theory of gravity, the latter appears to be about `twenty five percent wrong'. "
If "the latter" refers to "the Einstein's theory of gravity", then what's the "former"?
and so on.
Authors: We rephrase the sentence: Despite the enormous successes of Einstein's theory of gravity, it appears to be about ``twenty five percent wrong''.

Reviewer 3 Report
Comments and Suggestions for Authors
In the manuscript the authors investigate the galactic rotation curves by using the conformal gravity approach. A spherically symmetric solution is first presented, whose properties are studied in detail. Then the theoretical model is tested by fitting the observational data of 175 galaxies of the SPARC database. The manuscript is well written, and contains interesting results. Hence, I recommend the publication of the manuscript in its present form.
Comments on the Quality of English LanguageThe quality of English is good.
Author Response
Referee3: Comments and Suggestions for Authors
In the manuscript the authors investigate the galactic rotation curves by using the conformal gravity approach. A spherically symmetric solution is first presented, whose properties are studied in detail. Then the theoretical model is tested by fitting the observational data of 175 galaxies of the SPARC database. The manuscript is well written, and contains interesting results. Hence, I recommend the publication of the manuscript in its present form.
Comments on the Quality of English Language: The quality of English is good.
Authors: We thank the reviewer a lot for appreciating our research work.
Round 2
Reviewer 1 Report
Comments and Suggestions for Authors
I am satisfied with the corrections by the authors. Then I recommend the publication of this work.